# Multi-omics identify xanthine as a pro-survival metabolite for nematodes with mitochondrial dysfunction

Anna Gioran ⓘD, Antonia Piazzesi ⓘD, Fabio Bertan, Jonas Schroer, Lena Wischhof ⓘD, Pierluigi Nicotera ⓘD & Daniele Bano*ⓘD

## Abstract

Aberrant mitochondrial function contributes to the pathogenesis of various metabolic and chronic disorders. Inhibition of insulin/IGF-1 signaling (IIS) represents a promising avenue for the treatment of mitochondrial diseases, although many of the molecular mechanisms underlying this beneficial effect remain elusive. Using an unbiased multi-omics approach, we report here that IIS inhibition reduces protein synthesis and favors catabolism in mitochondrial deficient *Caenorhabditis elegans*. We unveil that the lifespan extension does not occur through the restoration of mitochondrial respiration, but as a consequence of an ATP-saving metabolic rewiring that is associated with an evolutionarily conserved phosphoproteome landscape. Furthermore, we identify xanthine accumulation as a prominent downstream metabolic output of IIS inhibition. We provide evidence that supplementation of FDA-approved xanthine derivatives is sufficient to promote fitness and survival of nematodes carrying mitochondrial lesions. Together, our data describe previously unknown molecular components of a metabolic network that can extend the lifespan of short-lived mitochondrial mutant animals.

**Keywords** insulin/IGF-1 signaling (IIS); metabolism; mitochondrial diseases; xanthine
**Subject Categories** Membrane & Intracellular Transport; Metabolism; Signal Transduction
**The EMBO Journal (2019) 38: e99558**

## Introduction

Mitochondria take part in key biological processes, supporting energy production as well as biosynthesis of various metabolic intermediates needed for cell growth and homeostasis. Aberrant mitochondrial bioenergetics is associated with a wide spectrum of age-related chronic diseases (Schon & Przedborski, 2011; Exner et al, 2012; Vafai & Mootha, 2012; Camandola & Mattson, 2017). Furthermore, it causally underlies several debilitating human pathologies commonly known as mitochondrial diseases (Koopman et al, 2012, 2016; DiMauro et al, 2013; Chinnery, 2015; Viscomi et al, 2015; Gorman et al, 2016). Representing the largest group of monogenic metabolic disorders, the most prominent hallmark of mitochondrial diseases is altered mitochondrial function and diminished oxidative phosphorylation (OXPHOS; Koopman et al, 2012; Gorman et al, 2016). A continuously growing number of mutations in nuclear as well as in mitochondrial genes have been linked to mitochondrial diseases (DiMauro et al, 2013; Turnbull & Rustin, 2016). These mutations may occur in a sporadic or inherited fashion (Gorman et al, 2016), and the resulting clinical manifestations show a striking variability in terms of age onset, organ involvement, symptoms, disease progression, and lifespan expectancy. Due to the genetic and structural complexity of the OXPHOS system (Koopman et al, 2012; Area-Gomez & Schon, 2014), the loose genotype–phenotype correlation complicates diagnostics as well as the development of effective treatments.

Currently, available therapeutic approaches are based on dietary supplements such as vitamins, free radical scavengers, and cofactors. Most of these agents are inadequate and mainly palliative for the wide spectrum of mitochondria-associated conditions. Nevertheless, remarkable progress has been made in the last years and newly proposed therapeutic interventions are now under evaluation in some multinational controlled clinical trials (Pfeffer et al, 2013; Viscomi et al, 2015; Koopman et al, 2016; Wang et al, 2016). Moreover, recent findings in experimental models suggest new concepts for beneficial metabolic alterations that may have important therapeutic implications. For instance, rebalancing of the $NAD^+$/NADH ratio through pharmacological interventions has been shown to improve the fitness of mice carrying mitochondrial defects (Canto et al, 2012; Cerutti et al, 2014; Pirinen et al, 2014). Moreover, brief exposure of complex I-deficient mice to hypoxic conditions activates transcriptional programs that promote compensatory biochemical pathways, overcoming the metabolic blockage and reducing the amount of potentially toxic byproducts due to impaired OXPHOS activity (Jain et al, 2016). Following the same concept of metabolic

German Center for Neurodegenerative Diseases (DZNE), Bonn, Germany
  *Corresponding author. Tel: +49 228 43302 510; Fax: +49 228 43302 689; E-mail: daniele.bano@dzne.de

rewiring, findings from independent laboratories demonstrated that genetic and pharmacological inhibition of the IIS pathway stimulates a metabolic response that seems to be beneficial in various mitochondrial disease models. In this regard, treatment with the mTOR inhibitor rapamycin extends the lifespan of complex I-deficient nematodes (Peng *et al*, 2015) and alleviates pathology in *Ndufs4* knockout, *Tk2* knockin, podocyte-specific *Phb2* knockout, mitochondrial DNA mutant, and AIF-deficient mice (Johnson *et al*, 2013; Ising *et al*, 2015; Khan *et al*, 2017; Siegmund *et al*, 2017; Wischhof *et al*, 2018). Moreover, rapamycin partially ameliorates kidney failure in coenzyme Q-deficient mutant mice (Peng *et al*, 2015) and maintains cellular ATP levels in human differentiated neurons exposed to OXPHOS inhibitors (Zheng *et al*, 2016). Together, this line of evidence suggests that decreased mTOR, and eventually Akt, pathway preserves energy and metabolite supply, prevents the accumulation of toxic species, and alleviates pathological features commonly associated with impaired mitochondrial function. Despite the importance of these findings, many molecular and cellular processes underlying these observed beneficial effects are not fully understood (Vafai & Mootha, 2013; Viscomi *et al*, 2015). In light of the above, we reasoned that unraveling these molecular mechanisms might provide valuable insight for therapeutic alternatives. Using a cross-species approach and unbiased methods, we explored the link between OXPHOS impairment, IIS, and the molecular cascades that counteract mitochondrial deficiency. Here, we show that genetic inhibition of the IIS pathway extends the survival of nematodes carrying mitochondrial lesions. Mechanistically, IIS inhibition stimulates a metabolic shift toward catabolic processes associated with xanthine buildup as a result of enhanced purine degradation. In cells and in nematodes carrying OXPHOS defects, xanthine derivatives can mimic several aspects of the IIS inhibition, such as activation of AMP-activated protein kinase (AMPK) and protein kinase A (PKA). As a proof of principle, we provide evidence that xanthine derivatives can counteract aberrant IIS due to mitochondrial lesions.

# Results

### Reduced IIS extends the survival of mitochondrial mutant nematodes

Several lines of evidence suggest that pharmacological modulation of IIS ameliorates pathological features associated with OXPHOS deficiency in mammals (Johnson *et al*, 2013; Ising *et al*, 2015; Khan *et al*, 2017; Siegmund *et al*, 2017; Wischhof *et al*, 2018). To identify molecular commonalities in *C. elegans*, we initially employed complex I-deficient nematodes as a model of OXPHOS impairment. Specifically, we used nematodes carrying a *gas-1* loss-of-function allele that was originally isolated in a screening of *C. elegans* mutant sensitive to volatile anesthetics (Kayser *et al*, 1999). The *gas-1* gene encodes for a respiratory complex I subunit that is homologous to the human NADH dehydrogenase iron-sulfur protein 2 (NDUFS2). The *gas-1(fc21)* allele causes the post-translational loss of complex I subunit NDUFS2 and, as a consequence, aberrant mitochondrial function, diminished fitness, altered neuronal dendritic outgrowth, and decreased survival (Hartman *et al*, 2001; Kayser *et al*, 2001; Ichishita *et al*, 2008; Gioran *et al*, 2014;

Troulinaki *et al*, 2018; Fig 1A–C and Dataset EV1). We extracted proteins from *gas-1(fc21)* and wild-type (wt) animals, performed proteomic analysis of isobarically labeled phospho-enriched peptides, and found over 900 differentially phosphorylated proteins (Fig 1D). Unbiased pathway analysis (i.e., Ingenuity Pathway Analysis, IPA) predicted several signaling pathways dysregulated in complex I-deficient nematodes (Appendix Fig S1A). Of the top upregulated signaling networks (Appendix Fig S1A), five pathways were predicted to converge and enhance IIS (Fig 1E), indicating that an aberrant IIS may be a shared feature of mitochondrial mutant nematodes and mammals. Given this, we set off to determine whether diminished IIS could promote survival of short-lived mitochondrial mutant nematodes. In *C. elegans*, the sole insulin/IGF-1 receptor DAF-2 regulates the activity of PI3K/AGE-1 and, through the downstream kinases PDK-1 and AKT-1/2, controls the transcription factor DAF-16 (Lin *et al*, 1997; Ogg *et al*, 1997; Kenyon, 2010). We generated *age-1(hx546); gas-1(fc21)* nematodes and observed that the aberrant phosphorylation status of the *gas-1(fc21)* proteome was almost completely dependent on IIS, as approximately 97% of these differentially phosphorylated proteins reverted to wt levels in *age-1(hx546); gas-1(fc21)* mutants (Appendix Fig S1B). Most importantly, we found that the lifespan of *age-1; gas-1* double mutant nematodes was significantly longer compared to *gas-1* mutants, as well as of wild-type (wt) and even *age-1(hx546)* single mutant animals (Fig 1B and Dataset EV1). In a similar manner, *pdk-1* loss of function extended *gas-1* lifespan (Appendix Fig S1C and Dataset EV1). Furthermore, we found that the lifespan extension depended primarily on the transcription factor DAF-16/FOXO, since *daf-16; age-1; gas-1* triple mutants lived significantly less than *age-1; gas-1* animals (Fig 1B and C, and Dataset EV1). It is worth noting that *daf-16; age-1; gas-1;* triple mutants did live a few days longer than *gas-1* single mutant as well as *daf-16; gas-1* double mutant animals, indicating that at least part of the lifespan-extending effects are not completely DAF-16-dependent (Fig 1C, Appendix Fig S1D, and Dataset EV1). Consistently, *daf-16* loss of function alone did not affect the *gas-1 (fc21)* survival (Appendix Fig S1D and Dataset EV1). Importantly, the lifespan-extending effect of IIS inhibition was not limited to complex I-deficient nematodes, since *age-1(hx546); mev-1(kn1)* double mutants also lived significantly longer compared to complex II-deficient *mev-1(kn1)* animals (Fig 1F and Dataset EV1). In the case of *mev-1* mutants, the increased survival was not as profound as in *gas-1*, possibly because of the more severe phenotype and stress sensitivity of complex II-deficient animals (see survival and censoring events in Dataset EV1). As an additional model, we used WAH-1/AIF-deficient nematodes, which we previously characterized for their decreased mitochondrial function and consequent reduced lifespan (Troulinaki *et al*, 2018). In line with our prior data, we found that IIS inhibition increased the lifespan of *wah-1*-silenced animals (Appendix Fig S1E and F and Dataset EV1).

We wondered whether the increased longevity of *age-1; gas-1* mutants could be due to improved mitochondrial function. We measured ATP and oxygen consumption rate (OCR) in *gas-1* and *age-1; gas-1* mutants. Compared to *gas-1* mutants, *age-1; gas-1* double mutant nematodes exhibited a large increase of ATP levels, despite the lack of improvement in basal mitochondrial respiration (Fig 1G and H). Intriguingly, we found that IIS inhibition improved

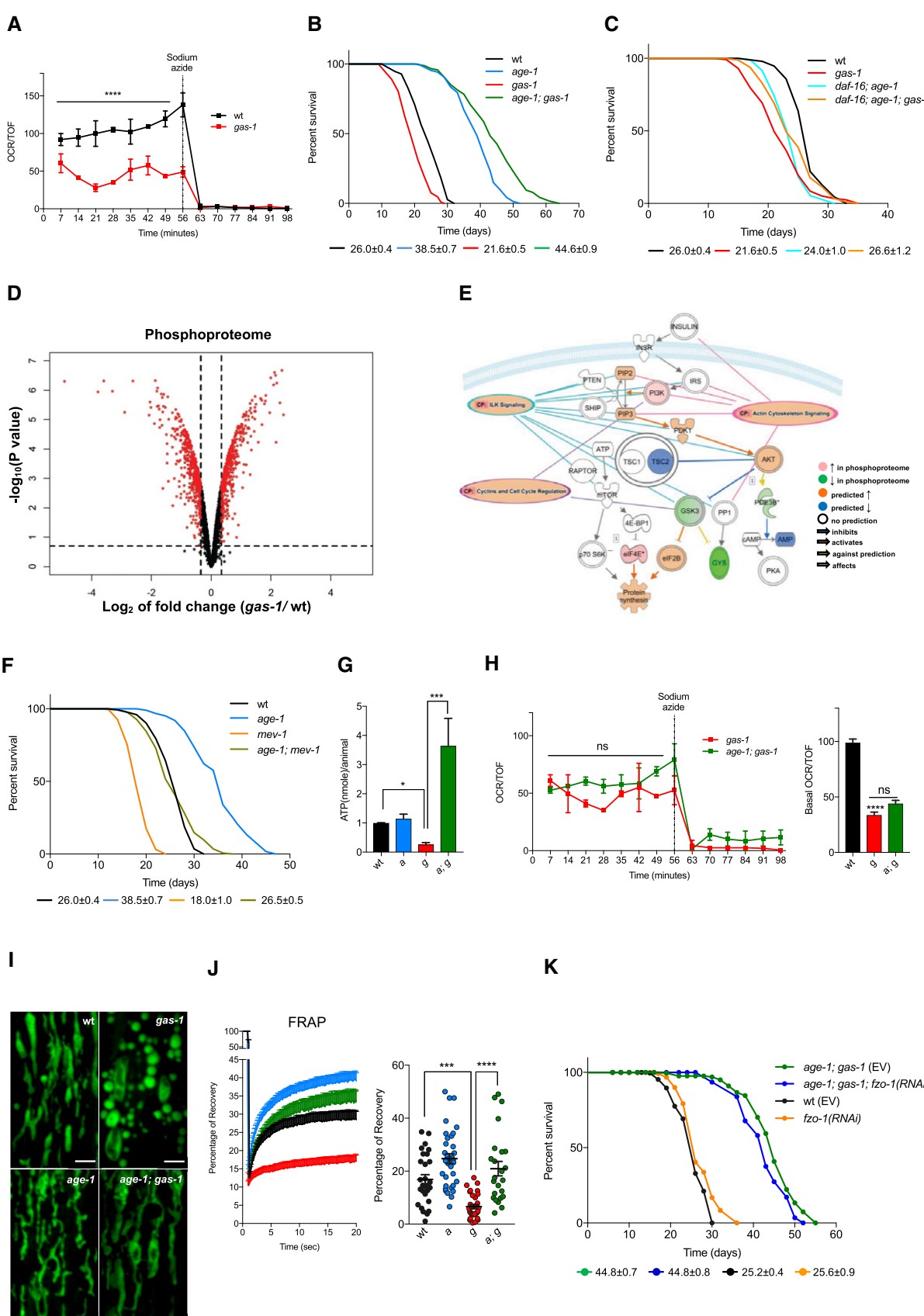

**Figure 1.**

**Figure 1.  Inhibition of IIS extends the lifespan of OXPHOS-deficient nematodes.**

A   OCR of wt and *gas-1* mutant nematodes [normalized to nematode size as given by the COPAS Biosort (Time of Flight—TOF)]. One representative curve is shown (mean ± SEM; *n* = 3) ****P-value < 0.0001; Kruskal–Wallis test, Dunn's *post hoc* correction.

B, C   Representative curves show the lifespan of wt and *gas-1(fc21)* nematodes compared to (B) *age-1* and *age-1; gas-1* mutants and (C) *daf-16* and *daf-16; age-1; gas-1* mutant animals. Average median lifespans ± SEM from all replicates indicated beneath representative curves.

D   Volcano plot of differentially phosphorylated proteins in *gas-1(fc21)* mutants compared to wt. Cutoffs: 20% FDR and |log$_2$(fold change)| > 0.345 (in red).

E   Schematic representation of IIS pathway including results and predictions from IPA of phosphoproteome of *gas-1(fc21)* compared to wt. Shades of red and green are proportional to fold change, as calculated by IPA.

F   Lifespan analysis of *meu-1(kn1)* nematodes compared to wt, *age-1(hx546)*, and *age-1(hx546); meu-1(kn1)* animals.

G   ATP measurements in wt, *age-1*, *gas-1*, and *age-1; gas-1* mutant nematodes (normalized to wt), *n* = 4. Bars: mean ± SEM, *P-value < 0.05, ***P-value = 0.001, Kruskal–Wallis test, Dunn's *post hoc* correction.

H   OCR of *gas-1* and *age-1; gas-1* mutant nematodes (normalized to TOF). Left panel: one representative curve (mean ± SEM). Right panel: basal respiration in wt, *gas-1*, and *age-1; gas-1* animals, pooled from three biological replicates, ****P-value < 0.0001, ns = non-significant, Kruskal–Wallis test, Dunn's *post hoc* correction.

I   STED images of representative mitochondrial morphology in muscle cells of L4 mutant worms. Scale bar: 2 μm.

J   Left panel: percentage of FRAP. Each point: mean recovery of ~30 bleached regions (2 regions per animal, 15 animals per condition) ± SEM. Right panel: values of the maximum (total) recovery per strain ± SEM, ***P-value < 0.001, ****P-value < 0.0001, one-way ANOVA, Tukey's *post hoc* correction.

K   Representative lifespan of *age-1; gas-1* nematodes fed with empty vector (EV) and *fzo-1* RNAi bacteria. Average median lifespans ± SEM from all replicates indicated beneath representative curves. For all panels: wt = wild type, *a = age-1*, *g = gas-1*, *a; g = age-1; gas-1*.

the maintenance of a more fused mitochondrial network as revealed by super-resolution STED microscopy and fluorescence recovery after photobleaching (FRAP) experiments in animals expressing a mitochondrial GFP in the muscle (Fig 1I and J, and Appendix Fig S1G). Since altered mitochondrial connectivity can influence *C. elegans* metabolism in several lifespan-extending paradigms (Yang *et al*, 2011; Chaudhari & Kipreos, 2017; Han *et al*, 2017; Weir *et al*, 2017), we reasoned that a more fused mitochondrial network may play a causative role for *age-1; gas-1* longevity. To test this hypothesis, we downregulated the mitochondrial fusion protein FZO-1. We found that *fzo-1* RNAi had a negligible effect on lifespan, despite the fact that it significantly fragmented the mitochondrial network (Fig 1K, Appendix Fig S1G–I, and Dataset EV1). Taken together, our findings suggest that IIS inhibition is associated with ATP maintenance, results in a more fused mitochondrial network, and promotes lifespan extension of mitochondria-deficient nematodes.

## IIS inhibition reduces ATP-consuming processes and enhances catabolism in mitochondrial mutant nematodes

To gain insight into the mechanisms underlying *age-1; gas-1* longevity, we performed mRNA next-generation sequencing (NGS) in wt, *gas-1* and *age-1; gas-1* mutant animals. A global analysis of the genome-wide expression profiling revealed that over 2,800 genes were significantly dysregulated in *gas-1* versus *age-1; gas-1* mutants (5% FDR). However, most of the dysregulated gene expression in *gas-1* mutants was not rescued upon IIS inhibition, with only approximately 13% of genes dysregulated in *gas-1* reverting back to wt levels in *age-1; gas-1* mutants (Fig 2A). To compare transcriptomic profiles from *gas-1* mutant and *age-1; gas-1* double mutant animals, we conducted an unbiased pathway analysis with IPA. We found that three of the most overrepresented pathways were involved in protein synthesis. Specifically, EIF2, eIF4/p70S6K and mTOR signaling pathways were significantly downregulated in *age-1; gas-1* compared to *gas-1* mutants (Fig 2B). To support the relevance of protein synthesis inhibition in longevity, we used a loss-of-function mutant for the translation initiation factor IFE-2/eIF4E, resulting in impaired mRNA translation initiation downstream TOR (Jankowska-Anyszka *et al*, 1998; Syntichaki *et al*, 2007). We observed that *ife-2; gas-1* double mutants had an increased lifespan compared to *gas-1* mutants, although they displayed a shorter survival compared to *age-1; gas-1* mutant nematodes (Fig 2C and Dataset EV1). Moreover, we found that *age-1; ife-2; gas-1* triple mutants lived approximately as long as *age-1; gas-1* double mutant nematodes (Fig 2C and

**Figure 2.  IIS inhibition affects the transcriptome and proteome of mitochondria-deficient nematodes.**

A   Left panel: Venn diagram of the number of genes significantly dysregulated in *age-1; gas-1* versus wt (green) and *gas-1* versus wt (red). Right panel: scatter plots of the expression of genes significantly downregulated (left) or upregulated (right) in *gas-1* and *age-1; gas-1* mutants compared to wt. In black: expression levels of genes that are significantly upregulated or downregulated in *gas-1* mutants compared to wt. In red: expression levels of the same genes in *age-1; gas-1* mutants compared to wt.

B   IPA indicating the overrepresented canonical pathways in *age-1; gas-1* versus *gas-1* mutants.

C   Representative lifespans of the indicated strains with average median lifespans ± SEM from all replicates indicated beneath representative curves.

D   Volcano plot of TMT/iTRAQ-based quantitative proteomic analysis. In red, proteins significantly dysregulated in *age-1; gas-1* compared to *gas-1* (FDR < 20%; |log$_2$ (fold change)| > 0.47). Dashed lines indicate cutoffs. Proteins involved in carbohydrate (green) and lipid (blue) metabolism are labeled.

E   STRING analysis on dysregulated proteins in *age-1; gas-1* versus *gas-1* nematodes (20% FDR; |log$_2$(fold change)| > 0.47).

F   IPA of TMT/iTRAQ scan results and prediction of downstream effects. Legend indicates color-coding of molecules and connections. Shades of red and green are proportional to fold change, whereas shades of blue and orange are proportional to the strength of the prediction, as calculated by IPA.

G   STRING network for proteins involved in carbohydrate (green) and lipid (blue) metabolism. Legend indicates the nature of each indicated connection.

H   Left panel: glucose quantification in wt, *gas-1*, and *age-1; gas-1* nematode lysates with NMR, *n* = 7, ns = non-significant, Mann–Whitney *U*-test. Right panel: quantification of total carbohydrates in the indicated strains with phenol–sulfuric assay. Bars: mean ± SEM, two biological replicates (five technical replicates each), normalized to wt and number of animals used. **P-value < 0.01, ns = non-significant, Mann–Whitney *U*-test.

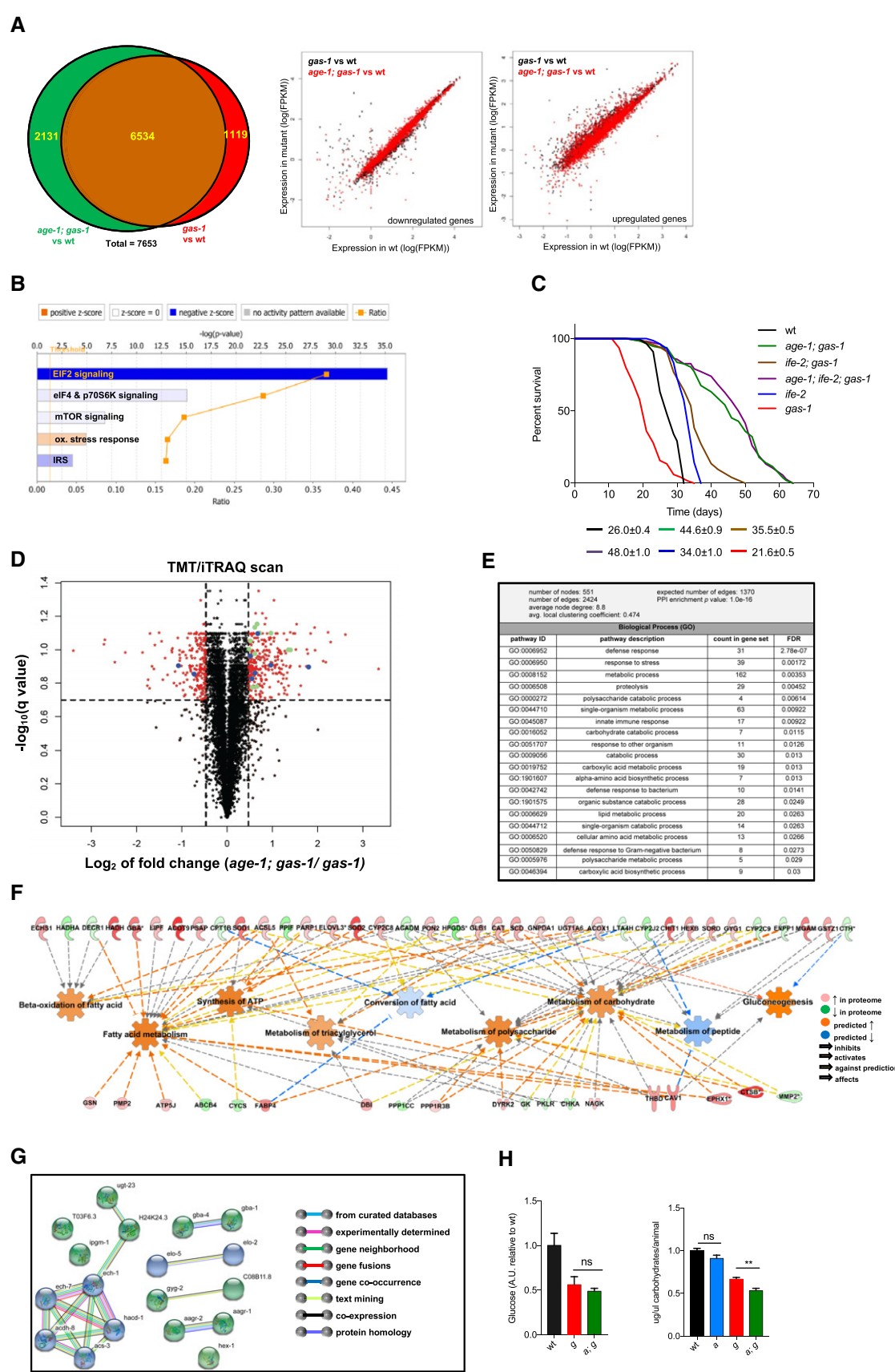

Figure 2.

Dataset EV1). Since *ife-2; gas-1* did not fully phenocopy the lifespan-extending effect of diminished IIS in *gas-1* animals, it is likely that inhibition of protein synthesis contributes only in part to *age-1; gas-1* survival. As a complementary approach to NGS, we obtained quantitative proteomic profiling using isobaric tags (TMT/iTRAQ) in *gas-1* and *age-1; gas-1* nematodes (Fig 2D). A protein–protein interaction network analysis (STRING, version 10.5; Szklarczyk *et al*, 2017) revealed that the hits of the proteomic scan formed networks significantly more than would be expected by chance (PPI enrichment *P*-value < 1.0e-16; Fig 2E and Appendix Fig S2A). IPA of the proteomic data predicted energy production (i.e., ATP synthesis) and lipid metabolism to be differentially regulated in *age-1; gas-1* compared to *gas-1* mutants (Fig 2F). Among these altered biological processes, we found fatty acid beta-oxidation and carbohydrate metabolism significantly represented (Fig 2F and G). Notably, a number of differentially regulated proteins were also significantly altered in the transcriptome, indicating that many of the proteome changes also occurred at the transcriptional level (Dataset EV2). Since genes in sugar metabolism seem particularly enriched in both the transcriptome and the proteome, we measured simple and complex carbohydrates using the phenol–sulfuric acid colorimetric assay. We observed a significant reduction in total carbohydrates; however, there were no differences in glucose levels between *age-1; gas-1* mutant and *gas-1* mutant animals (Fig 2H). We then overlapped IPA predictions from both transcriptomics and proteomics. In *age-1; gas-1* compared to *gas-1* mutant animals, we found that fatty acid β-oxidation, gluconeogenesis, and glycolysis were all upregulated, whereas glycogen turnover (i.e., lysis and synthesis) was highly enhanced (Appendix Fig S2B). Based on this pathway analysis, it seems that IIS inhibition supports the maintenance of glucose through the engagement of multiple gluconeogenic processes. Consistent with this hypothesis and with our transcriptome, we found that the expression of two enzymes involved in carbohydrate metabolism, the alpha-glucosidases AAGR-1 and AAGR-2, were upregulated on mRNA level in *age-1; gas-1*

compared to *gas-1* (Appendix Fig S2C). Moreover, the expression of the two *C. elegans* orthologs of phosphoenolpyruvate carboxykinase (PEPCK) was equally altered in *age-1; gas-1* compared to *gas-1* (Appendix Fig S2D).

Next, we sought to identify the molecular players underlying these metabolic changes in *age-1; gas-1* mutant animals. A further IPA of our NGS data supported the activation of AMP-activated protein kinase (AAK-2/AMPK) and protein kinase A (KIN-1/PKA) and the consequent effect on distinct catabolic pathways, including fatty acid beta-oxidation and glycogen metabolism in *age-1; gas-1* double mutants compared to *gas-1* animals (Fig 3A and B). In line with its role in conditions of low energy and nutrient deprivation (Burkewitz *et al*, 2014; Hardie, 2014) and with our IPA predictions, we found that *age-1; gas-1* double mutants had increased levels of phospho-AAK-2 compared to *gas-1* animals (Fig 3C) as revealed using a previously validated antibody (Gioran *et al*, 2014). Importantly, *aak-2* loss of function and silencing significantly reduced the lifespan of *age-1; gas-1* double mutants, whereas overexpression of AAK-2 promoted the survival of *gas-1* mutants (Fig 3D and E, Appendix Fig S3A and B, and Dataset EV1). In *C. elegans*, *kin-1* encodes the catalytic subunit of PKA, whereas *kin-2* encodes for the regulatory one (Lee *et al*, 2016). In line with the IPA prediction, we found that *age-1; gas-1* double mutants displayed an increased KIN-1 activity compared to *gas-1* mutant animals (Fig 3F). Furthermore, RNAi-mediated downregulation of *kin-1* almost completely abolished the lifespan extension of *age-1; gas-1* animals (Fig 3G, Appendix Fig S3C and D, and Dataset EV1). Moreover, downregulation of *aak-2* and *kin-1* affected ATP levels in *age-1; gas-1* double mutants (Fig 3H and I). Notably, we observed that the mitochondrial network was significantly more fragmented in *aak-2-* and *kin-1*-deficient animals (Fig 3J), suggesting a mechanistic link between mitochondrial network remodeling and the kinases involved in lifespan-extending processes. Taken together, IIS inhibition decreases ATP-consuming processes in complex I-deficient nematodes and sustains AMPK/AAK-2 and PKA/KIN-1/2 activity.

---

**Figure 3.   Increased AMPK/AAK-2 and PKA/KIN-1/2 activity occur in the long-lived *age-1; gas-1* mutants.**

A, B  Schematic representation with overlaid Molecule Activity Predictor (MAP) analysis of (A) AAK-2/AMPK- and (B) KIN-1/PKA-regulated metabolic processes in *age-1; gas-1* versus *gas-1* mutants. Color intensity corresponds to effect intensity or prediction strength.

C  Immunoblot analysis of phospho-AAK-2 in wt, *age-1*, *gas-1*, and *age-1; gas-1* mutant nematodes (actin loading control). Numbers correspond to band densitometries across three biological replicates.

D  Representative lifespan of *age-1; gas-1* double mutant and *age-1; aak-2; gas-1* triple mutant nematodes with average median lifespans ± SEM from all replicates indicated beneath representative curves.

E  Lifespan assay of *gas-1* mutant animals compared to *gas-1*; AAK-2-overexpressing (O/E) nematodes and their wt counterparts. Inset: immunoblot analysis of phospho-AAK-2 in *gas-1* (*g*) and *gas-1*; AAK-2 O/E (*g; a O/E*) animals (actin loading control), showing the overexpressed protein (lower band) and the endogenous one (upper band).

F  Left panel: immunoblot analysis of phospho-KIN-1 substrates in young adult nematodes. Numbers correspond to band densitometries across two biological replicates. Right panel: immunoblot analysis of phospho-KIN-1 substrates in *age-1; gas-1* animals grown on bacteria carrying an empty vector (EV) or expressing *kin-1* RNAi. Two different *kin-1* RNAi clones (g1, b5) were used in this experiment. Numbers correspond to band densitometries of the presented immunoblot. Dashed rectangles: band corresponding to downregulated KIN-1 phospho-substrate.

G  Representative lifespan assay of wt and *age-1; gas-1* nematodes fed with empty vector (EV) and *kin-1* RNAi bacteria with average median lifespans ± SEM from all replicates indicated beneath representative curves.

H, I  ATP measurements in: (H) *age-1(hx546); gas-1(fc21)* (*a; g*) and *age-1(hx546); aak-2(ok524); gas-1(fc21)* (*a; aa; g*) mutant nematodes; (I) *age-1; gas-1* (EV) (*a; g* (EV)) and *age-1; gas-1; kin-1 (RNAi)* (*a; g; kin-1(RNAi)*). Bars: mean ± SEM, *n* = 4, *P*-value < 0.05, Mann–Whitney *U*-test.

J  Left panel: curves represent FRAP in *age-1; gas-1* animals grown on RNAi from larval L1 (or from egg for *fzo-1* RNAi) to L4 stage. Each point: mean recovery of bleached regions (2–5 regions per animal, 10–20 animals per condition) ± SEM. Right panel: maximum (total) recovery per condition are shown, ±SEM. *P*-value < 0.05, **P*-value < 0.01, ****P*-value < 0.0001, Kruskal–Wallis test, Dunn's *post hoc* correction. For all lifespan curves, the legend shows average median lifespan ± SEM across all biological replicates.

Source data are available online for this figure.

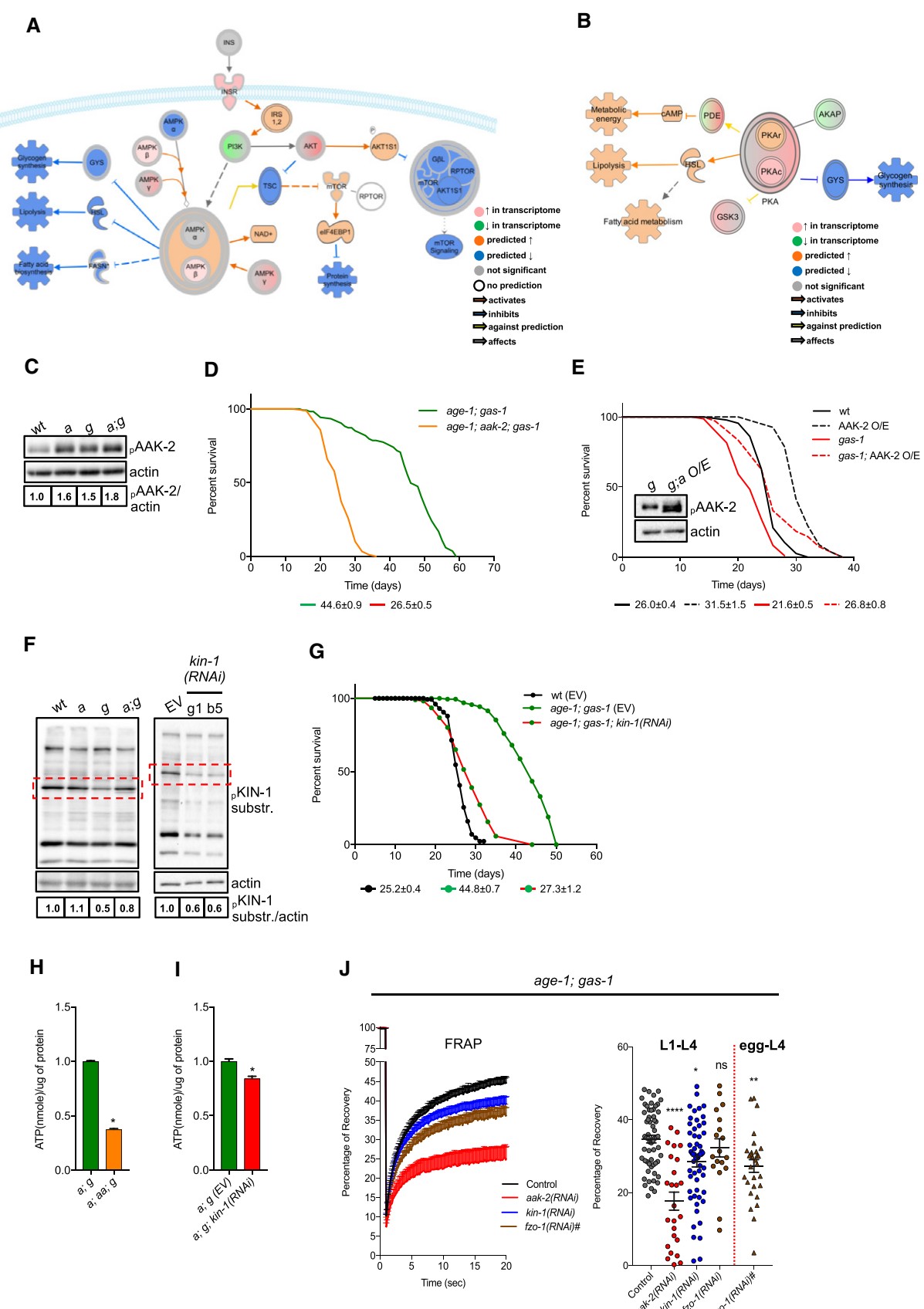

**Figure 3.**

## IIS inhibition induces nucleotide breakdown and xanthine accumulation in mitochondrial mutant nematodes

We reasoned that enhanced catabolism would result in different metabolic profiles in *age-1; gas-1* double mutant compared to *gas-1* mutant animals. First, we performed a lipid analysis and, to our surprise, found that the lipid profiling showed only minor differences between *gas-1* and *age-1; gas-1* mutants, with no significant reduction of the storage lipid triacylglycerols (TAGs; Fig 4A). Then, we assessed common primary metabolic intermediates in our mitochondrial mutant nematodes. On the metabolite level, an NMR targeted profiling on nematode lysates showed only few changes, with an almost sixfold increase in xanthine being the most prominent alteration in *age-1; gas-1* compared to *gas-1* mutant animals (Fig 4B). Based on the IPA of the transcriptomic data, it seems that the observed xanthine buildup is a consequence of increased purine degradation in *age-1; gas-1* animals (Fig 4C). Consistently, qRT–PCR showed that xanthine dehydrogenase *xdh-1* was significantly reduced in *age-1; gas-1* compared to *gas-1* mutant animals, while xanthine oxidase activity was not significantly affected (Fig 4D). These data point toward a scenario in which xanthine buildup occurs through inhibition of xanthine dehydrogenase (i.e., xanthine degradation) and not through activation of xanthine oxidase (i.e., xanthine synthesis).

We reasoned that xanthine buildup might be part of a molecular cascade that limits the detrimental effect of mitochondrial dysfunction. This hypothesis raises intriguing possibilities in terms of therapeutic options, as xanthine derivatives are relatively well tolerated and completely metabolized by humans (Aviado & Porter, 1984). To investigate the effect of xanthine derivatives on mitochondrial mutant cells, we used genetically modified HAP1 cells lacking mitochondrial complex I subunits *NDUFS2*, *NDUFS4*, or *NDUFA9* (Appendix Fig S4A). These tumorigenic cells showed impaired mitochondrial respiration (Appendix Fig S4B) and enhanced IIS as revealed by the increased phosphorylation of mTOR and Akt targets (i.e., pAkt$^{Ser473}$ and pPRAS40$^{Thr246}$, respectively; Appendix Fig S4C). Given the low solubility of xanthine in aqueous solutions, we employed aminophylline (AMP), enprofylline (ENPF), and pentoxifylline (PTX; Fig 4E), which are three widely used methylxanthines generally known as competitive, nonselective phosphodiesterase inhibitors (PDEs) and adenosine receptor antagonists (Snyder *et al*, 1981; Monteiro *et al*, 2016). All three methylxanthines partially attenuated the phosphorylation status of Akt and mTOR targets

(Fig 4F) and enhanced PKA activity (Fig 4G). This effect on IIS was not a common feature of PDE inhibitors, since the PDE3 inhibitor cilostazol enhanced PKA activity without altering the phosphorylation status of Akt (Appendix Fig S4D). In the range of concentrations that we used, methylxanthines may act as adenosine receptor antagonists (Snyder *et al*, 1981; Monteiro *et al*, 2016). However, since adenosine receptor antagonists MSX-3 and SCH58261 had no effect on either PKA or Akt phosphorylation (Appendix Fig S4E), it is likely that methylxanthines acted on PKA and Akt phosphorylation independently of adenosine receptor antagonism. Notably, treatment with PTX (Appendix Fig S4F) and other methylxanthines (data not shown) did not improve mitochondrial respiration.

Having established the pleiotropic effects of xanthine derivatives on Akt/mTOR and PKA activity, we sought to determine whether increased xanthine levels had lifespan-extending properties. Thus, we exposed wt and *gas-1* mutant nematodes to xanthine derivatives by supplementing them with the bacteria. While xanthine derivatives did not affect the lifespan of wt animals (Fig 5A), we found that these three methylxanthines significantly improved locomotor activity and extended the survival of complex I-deficient *gas-1* mutants (Fig 5B and C, and Dataset EV1). The lifespan-extending effect of methylxanthines was not due to decreased food intake, since compound-treated animals showed a significantly increased pharyngeal pumping (Fig 5D). We also found that KIN-1/2 activity and AAK-2 phosphorylation were enhanced in methylxanthine-treated *gas-1* mutant nematodes (Fig 5E and F). Notably, PTX treatment promoted mitochondrial connectivity in *gas-1* mutant nematodes (Fig 5G), phenocopying in part what we observed in *age-1; gas-1* double mutants (Fig 1I and J). Taken together, supplementation of xanthine derivatives enhances AMPK/AAK-2 and PKA/KIN-1 activity, promotes mitochondrial network remodeling, and increases survival of complex I-deficient nematodes (Fig 5H). To a certain extent, IIS inhibition and treatment with methylxanthines share common molecular processes that underlie the increased survival of short-lived mitochondrial mutants.

## Discussion

Over the past years, the remarkable progress of molecular medicine has broadened our understanding of mitochondrial function and its

**Figure 4.  Xanthine accumulates in *age-1; gas-1* mutants.**

A  Volcano plot of lipidomic profiling in *gas-1* versus *age-1; gas-1* mutants. Red stars: significantly altered lipids in *age-1; gas-1* versus *gas-1* (5% FDR; *n* = 5).

B  NMR-1D targeted metabolic profiling in *age-1; gas-1* mutants (normalized to *gas-1*). Dashed lines indicate fold change cutoff (|log$_2$(fold change)| < 1.23). Red bars: not significant, Light blue bars: FDR < 10%, |log$_2$(fold change)| < 1.23, Dark blue bars: FDR < 10%, |log$_2$(fold change)| > 1.23. Bar: mean + SEM, *n* = 7.

C  Schematic representation of purine degradation pathway with overlaid Molecule Activity Predictor (MAP) analysis showing increased purine degradation in *age-1; gas-1* mutants and the consequent xanthine accumulation. Shades of red and green are proportional to fold change, whereas shades of blue and orange are proportional to the strength of the prediction, as calculated by IPA.

D  Left panel: qRT–PCR of *xdh-1* (normalized to wt). Bars: mean ± SEM, *n* = 6. ****P-value < 0.0001, *P-value < 0.05, ns = non-significant, Kruskal–Wallis with Dunn's *post hoc* test. Right: xanthine oxidase activity levels normalized to animal number. Bars: mean ± SEM, *n* = 2, ns = non-significant, Mann–Whitney *U*-test.

E  Schematic representations of xanthine and the methylxanthines used in this study.

F  Immunoblot analysis of phospho-Akt (S473), pan Akt, phospho-PRAS40 (T246), and total PRAS40 in WT and *NDUFA9* KO HAP1 cells treated with 0.2% DMSO (−) or AMP (2 μM), ENPF (2 μM), and PTX (36 μM) (+) for 24 h (actin loading control).

G  Immunoblot analysis of phospho-PKA substrates in WT and *NDUFA9* KO HAP1 cells treated as indicated (actin loading control). For all immunoblot panels, numbers correspond to band densitometries across 2–3 biological replicates.

Source data are available online for this figure.

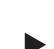

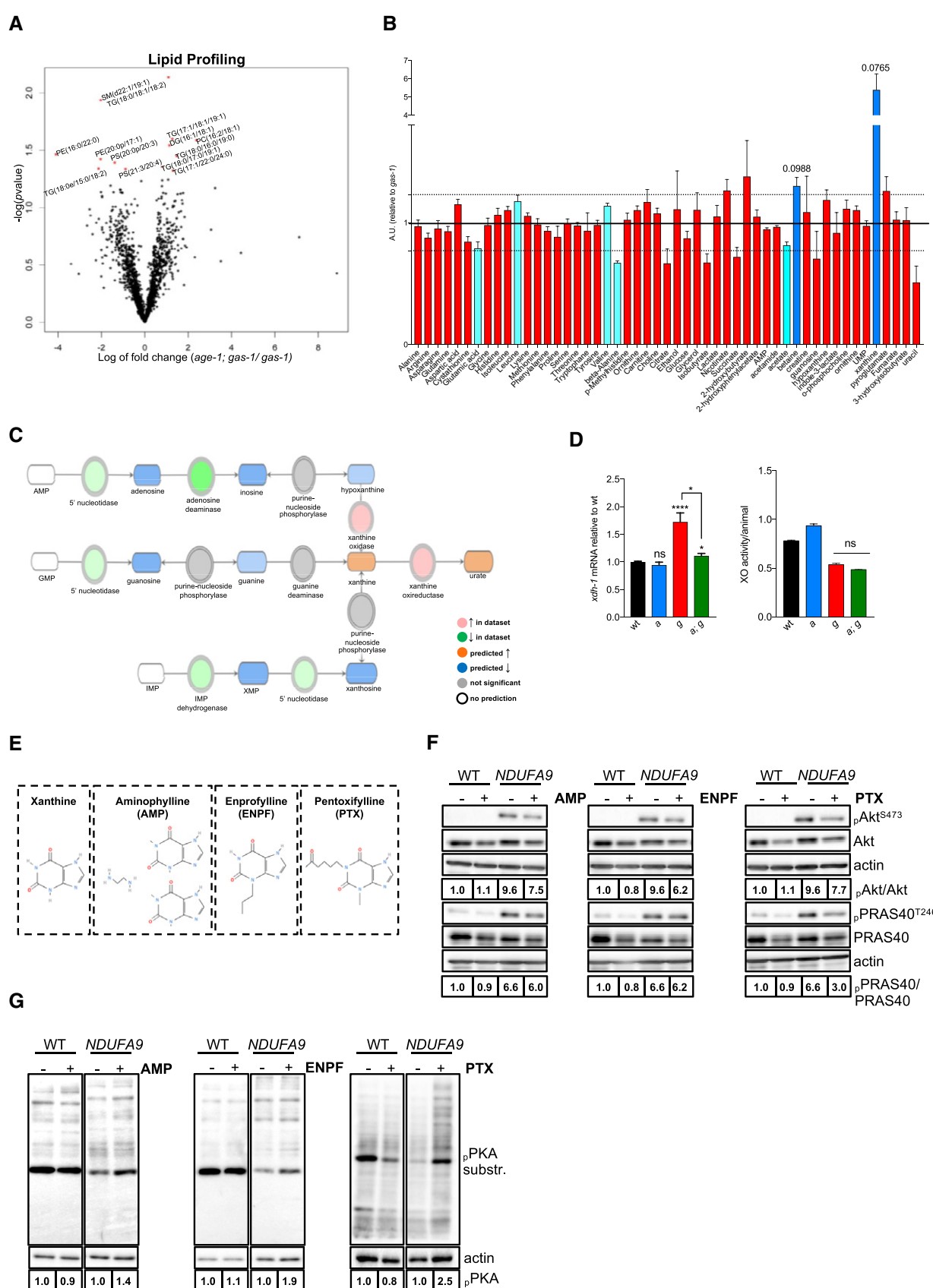

**Figure 4.**

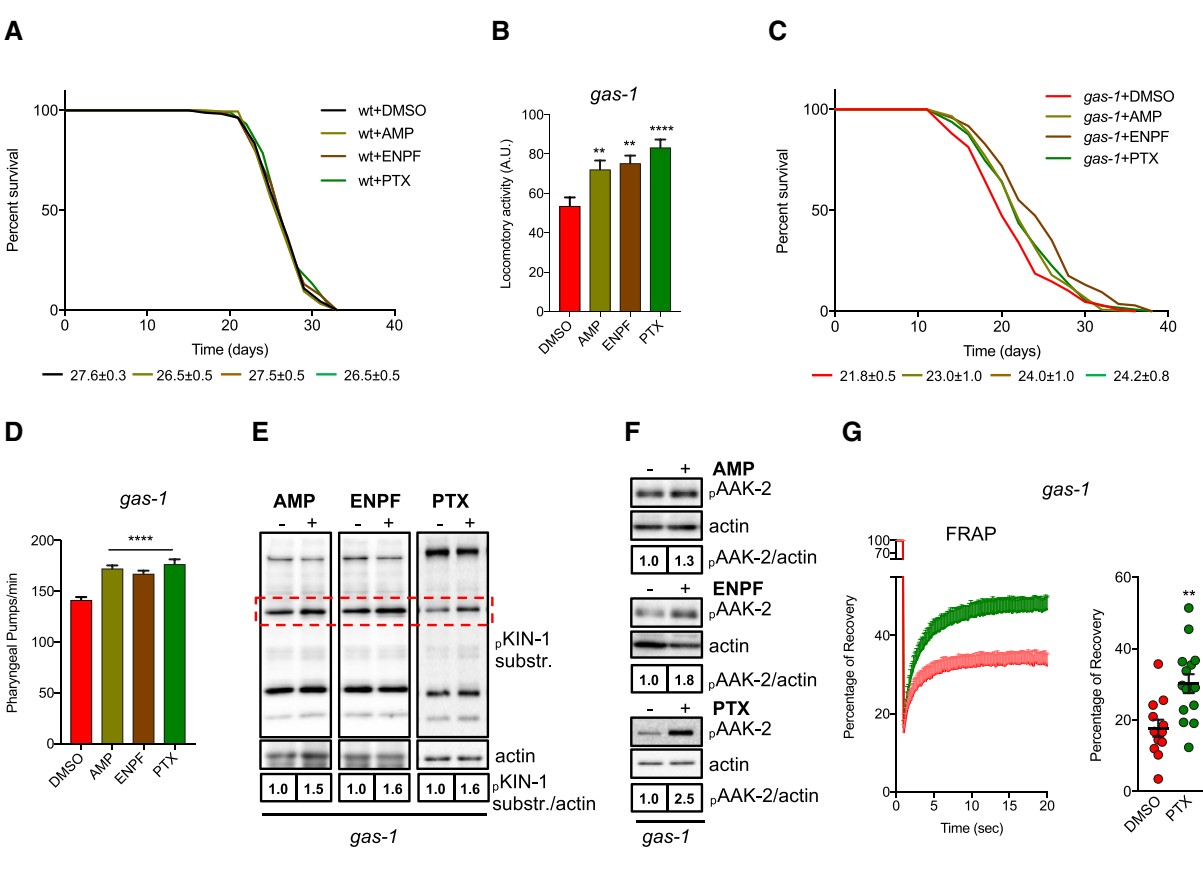

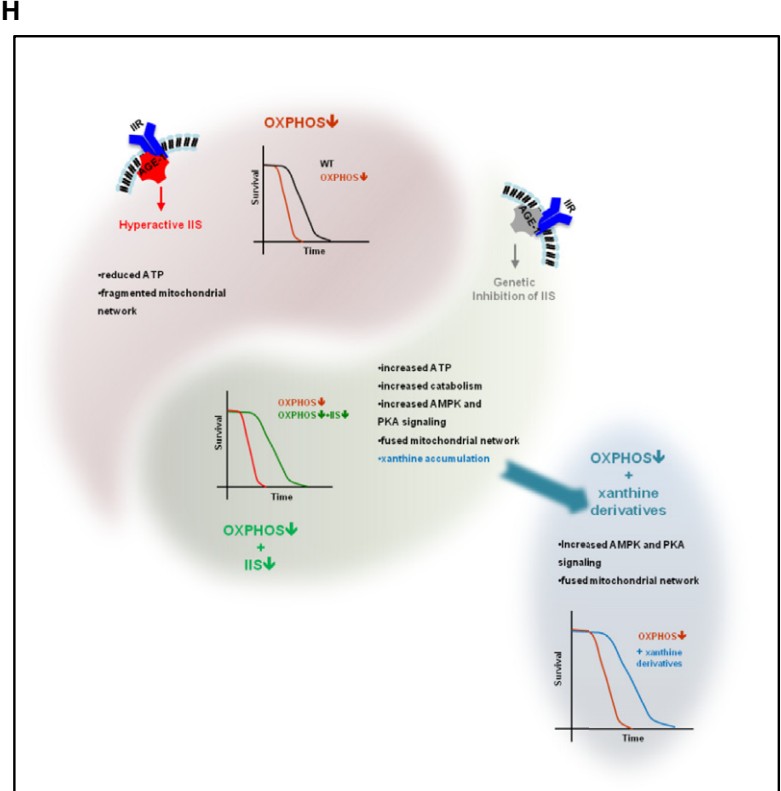

Figure 5.

**Figure 5.  Methylxanthines have lifespan-extending effects in *gas-1* mutant nematodes.**

A   Effect of methylxanthines on wt lifespan. Average median lifespans ± SEM from all replicates indicated beneath representative curve.

B   Locomotor activity of 10-day-old *gas-1* mutant nematodes treated with vehicle (DMSO) or methylxanthines. Values represent arbitrary units from three biological replicates with four technical replicates each, 35 worms per technical replicate. Bars: mean ± SEM, **P-value < 0.01, ***P-value < 0.001, one-way ANOVA, Dunnett's *post hoc* correction.

C   Effect of methylxanthines on the survival of *gas-1* mutants. Average median lifespans ± SEM from all replicates indicated beneath representative curves.

D   Pharyngeal pumps/minute in young adult *gas-1* mutant nematodes treated with vehicle or methylxanthines. Bars: mean ± SEM, n = 18. ****P-value < 0.0001, one-way ANOVA, Dunnett's *post hoc* correction.

E, F   Immunoblot analysis of (E) KIN-1 phospho-substrates and (F) phospho-AAK-2 in young adult *gas-1* mutant nematodes treated with vehicle (DMSO) or methylxanthines (actin loading control).

G   Left panel: FRAP in vehicle or PTX-treated *gas-1* animals. Each point: mean recovery of 14 bleached regions (2 regions per animal, 7 animals per condition) ± SEM. Right panel: maximum (total) recovery per condition ± SEM. **P-value < 0.01 unpaired *t*-test. For all immunoblot panels, numbers correspond to band densitometries across biological replicates, and for all lifespan curves, the legend shows mean–median lifespan ± SEM across all biological replicates.

H   Schematic drawing showing the critical biological processes underlying the lifespan extension of *age-1; gas-1* mutant nematodes. OXPHOS-deficient nematodes display decreased lifespan, diminished ATP production, fragmented mitochondrial network, and a phosphoproteome signature associated with enhanced IIS signaling. As a result of IIS inhibition, AMPK and PKA drive a catabolic shift underlying lifespan extension of nematodes carrying OXPHOS defects. Supplementation of xanthine derivatives stimulates survival of mitochondrial mutant animals.

Source data are available online for this figure.

contribution to a large range of human pathologies, including metabolic and age-related neurodegenerative disorders (Schon & Przedborski, 2011; Koopman *et al*, 2012; Camandola & Mattson, 2017). Mitochondrial diseases are among the most common forms of inherited myopathies and neurological conditions when considered as a broad group and independently from their etiology (DiMauro *et al*, 2013; Area-Gomez & Schon, 2014; Gorman *et al*, 2016). From a therapeutic standpoint, it would be ideal if one therapeutic approach could target the wide spectrum of mitochondria-associated diseases, considering that many of them share pathogenic signaling cascades (Avula *et al*, 2014; Viscomi *et al*, 2015; Wang *et al*, 2016). Surely, the repurposing of an FDA-approved molecule, like rapamycin, has many advantages, since pre-existing clinical data and a well-defined toxicity profile in humans are available (Laplante & Sabatini, 2012; Vafai & Mootha, 2013; Li *et al*, 2014; Kennedy & Lamming, 2016). However, long-term administration of rapamycin has troubling caveats. For example, it is known that rapamycin possesses immunosuppressive properties, may accelerate the growth of certain solid tumors, and induces insulin resistance, hypertriglyceridemia, and hyperglycemia (Meric-Bernstam & Gonzalez-Angulo, 2009; Laplante & Sabatini, 2012; Li *et al*, 2014). Given this and in the hope of finding alternative compounds and/or targets for therapeutic interventions, we investigated how genetic inhibition of the IIS has lifespan-extending properties in OXPHOS-deficient organisms. To do so, we employed an unbiased comparative approach and conducted a complementary set of quantitative methods that included phosphoproteomics, transcriptomics, proteomics, lipidomics, metabolomics, and pathway analysis. As part of our target validation, we then defined the epistatic relationship between the IIS cascade and novel molecular components predicted to participate in the metabolic reprogramming of mitochondrial mutant nematodes. We found that genetic IIS inhibition promotes the survival and fitness of mitochondria-deficient animals, likely attributable to the boosted ATP levels despite having severe genetic lesions. Reasoning that this could be due to a recovery of mitochondrial function, we observed instead no improvement in oxygen consumption rate. We found that genetic IIS inhibition leads to a remarkably more fused mitochondrial network that is only associated with, rather than being causally linked to lifespan extension. We are indeed surprised that *fzo-1* RNAi does not alter *age-1; gas-1* survival, since prior evidence shows that a fused mitochondrial network is required for the establishment of distinct pro-longevity programs (Yang *et al*, 2011; Chaudhari & Kipreos, 2017; Weir *et al*, 2017). Our findings do not argue against the lifespan-extending properties of mitochondrial network remodeling. Rather, we believe that, in *age-1; gas-1* animals, enhanced mitochondrial connectivity is simply the consequence of a homeostatic response, which optimizes the usage of nutrients and restores the balance in a bioenergetically compromised context where mitochondrial ATP production is insufficient. In line with this hypothesis, our transcriptomics, proteomics, and genetic epistatic analyses indicate that IIS inhibition efficiently decreases ATP-consuming anabolic processes (i.e., protein synthesis) and enhances catabolism (e.g., glycogen degradation and fatty acid beta-oxidation) in animals carrying mitochondrial lesions.

The connection between mitochondrial deficiency and IIS has been extensively investigated in mammals (Johnson *et al*, 2013; Ising *et al*, 2015; Khan *et al*, 2017; Siegmund *et al*, 2017; Wischhof *et al*, 2018). As part of our study, we provide a comprehensive overview of the phosphoproteome landscape in *gas-1* mutants. It clearly indicates that IIS arguably has the most profound impact on protein phosphorylation in complex I-deficient nematodes. Not only did IPA reveal that five of the seven most hyperactive pathways converge on IIS, but also that 97% of significantly affected phosphorylated proteins in *gas-1* mutants were found to revert back to wt levels in *age-1; gas-1* double mutants. In fact, there were very few differentially phosphorylated proteins between wt and *age-1; gas-1* mutants (data not shown), which further strengthened the hypothesis that the inhibition of IIS may counteract mitochondrial deficiency. What was perhaps most surprising was the degree of the lifespan extension observed in *age-1; gas-1* mutants. Far from simply returning their median lifespan to wild-type levels, as might have been inferred from the phosphoproteome, complex I-deficient nematodes actually lived even longer than wt animals upon IIS inhibition. Based on our findings, it seems reasonable to think that such a lifespan extension is the result of a complex interplay between different metabolic networks. In line with this argument, the lifespan-extending effect of IIS inhibition critically depends on AMPK and PKA, since their downregulation drastically affects the survival of *age-1; gas-1* mutants. Based on a statistical analysis performed by

IPA, it seems that these two pleiotropic kinases form a signaling module upstream of the route associated with metabolic flexibility and shift toward catabolism. While we focused on AMPK and PKA and confirmed that they are two critical regulators of these pro-survival processes, we do not rule out that additional components (e.g., kinases, phosphatases) of this essential metabolic node may participate in the aforementioned adaptive response.

We identified enhanced purine degradation and the subsequent xanthine buildup as a main downstream event that occurs in mitochondria-deficient nematodes upon IIS inhibition. We provide evidence that xanthine derivatives have pleiotropic effects in complex I-deficient cells, since they attenuate hyperactive IIS and stimulate PKA activity. This is not the first study to connect xanthine and IIS inhibition in *C. elegans*. Previously, xanthine along with adenine has been found to be slightly, yet significantly reduced in IIS-deficient nematodes within a larger context of metabolic flexibility (Gao *et al*, 2018). In contrast, our unbiased approach unveils a massive increase in xanthine levels in IIS-deficient nematodes carrying mitochondrial defects. Although our work leaves open the possibility that methylxanthines may have additional targets in nematodes, we clearly show that treatment with these xanthine derivatives is enough to significantly promote survival of complex I-deficient *gas-1* mutants, despite being a much milder effect than a hypomorphic mutation that drastically affects IIS. Thus, it seems that the accumulation of xanthine is not incidental, but rather it provides a specific feedback signal that establishes metabolic flexibility in animals carrying mitochondrial lesions. The lack of an "anti-aging" effect in wild-type animals supports this hypothesis. Finally, it is worth noting that methylxanthines stimulate AAK-2/AMPK and KIN-1/PKA activity and promote a more fused mitochondrial network in *gas-1* mutants, mimicking in part the genetic effect of IIS inhibition. Taken together, our study identifies molecular components that contribute to the survival of IIS-deficient animals carrying mitochondrial lesions. As an additional value, we show that methylxanthines can attenuate hyperactive IIS in OXPHOS impaired cells, as well as have positive effects on the survival of short-lived *C. elegans* mitochondrial mutants. Finally, our findings may hold possible translational implications, since novel molecular processes here described as well as newly modified methylxanthines could be considered in the fight against diseases associated with mitochondrial dysfunction.

# Materials and Methods

## Antibody list

The following antibodies were used in this study: α-pAkt$^{S473}$ (#4058), α-Akt(pan) (#4685), α-pPRAS40$^{T246}$ (#13175), α-PRAS40 (#2691), α-phospho-PKA substrates (#9624), α-pAMPK$^{T172}$ (#2535) (Cell Signaling Technology). Mouse monoclonal antibodies against mammalian and nematode actin were obtained from Sigma and Abcam, respectively. Secondary α-mouse and α-rabbit antibodies were obtained from Invitrogen and Pierce, respectively. All primary antibodies were used at 1:1,000 in TBST (Tris-buffered saline + 0.1% Tween-20), except for α-actin (1:5,000). Secondary antibodies were used at 1:2,000 in TBST.

## ATP measurements

ATP was measured using a Colorimetric Assay Kit (Sigma, MAK190). Young adult nematodes were sorted and collected with a COPAS Biosort System in pellets, flash-frozen in liquid $N_2$, and stored at −80°C until use. All pellets were resuspended in ATP assay buffer and disrupted with a sonicator. The deproteinization step was conducted with 10 kDa cutoff spin columns (Abcam) in a chilled microcentrifuge for 1 h at 9,600 *g*. Absorbance was measured with a Multiskan FC Plate Reader (Thermo Fisher Scientific) at 570 nm. For each sample, ATP concentration was normalized to animal number for wt, *age-1*, *gas-1* and *age-1; gas-1* nematodes) or protein content (for *age-1; gas-1*, *age-1; aak-2; gas-1*, *age-1; gas-1* (EV), and *age-1; gas-1; kin-1(RNAi)* nematodes).

## *C. elegans* strains and methods

The following *C. elegans* strains were used in this work: wild-type N2 (Bristol), AGD731 *uthEx299[aak-2(genomic aa1-aa321)::GFP:: unc-54 3′UTR + myo-2p::tdTomato]*, BAN1 *daf-2(e1370)III* (7× backcrossed), BAN57 *daf-16(mu86)I; daf-2(e1370)III; gas-1(fc21)X*, BAN153 *gas-1(fc21)X; zcIs14[myo-3p::GFP(mit)]*, BAN201 *ife-2 (ok306)X; gas-1(fc21)X*, BAN202 *age-1(hx546)II; gas-1(fc21)X*, BAN203 *gas-1(fc21)X; uthEx299[aak-2(genomic aa1-aa321)::gfp:: unc-54 3′UTR + myo-2p::tdTomato]*, BAN232 *age-1(hx546)II; ife-2 (ok306)X; gas-1(fc21)X*, BAN264 *age-1(hx546)II; zcIs14[myo-3p:: GFP(mit)]*, BAN267 *age-1(hx546)II; gas-1(fc21)X; zcIs14[myo-3p:: GFP(mit)]*, BAN285 *age-1(hx546)II; mev-1(kn1)III*, BAN286 *daf-16 (mu86)I; age-1(hx546)II; gas-1(fc21)X*, BAN287 *daf-16(mu86)I; age-1(hx546)II*, BAN308 *age-1(hx546)II; aak-2(ok524)X; gas-1(fc21)X*, BAN323 *pdk-1(mg142)X; gas-1(fc21)X*, BAN324 *daf-16(mu86)I; gas-1(fc21)X*, CW152 *gas-1(fc21)X*, GR1318 *pdk-1(mg142)X*, RB579 *ife-2 (ok306)X*, SJ4103 *zcIs14[myo-3p::GFP(mit)]*, TJ1052 *age-1(hx546)II, daf-16(mu86)I; daf-2(e1370)III*. All experiments were conducted at 20°C with isogenic hermaphrodite nematode populations. For the lifespan assays, synchronized day-4 (since hatching) young adults were transferred to fresh NGM plates seeded with OP50, RNAi, or OP50 mixed with compounds every other day until they were past the egg-laying stage. After that, transfers were done twice per week and all nematodes were scored every other day. Animals that did not respond to gentle prodding with a platinum wire were scored as dead. Animals that died during the experiment due to internal egg hatching, vulva protrusion or dried on the edges of the plates were scored as censored. qRT–PCR (see below) was used to assess the RNAi efficiency on nematodes grown on RNAi-expressing bacteria for 3–5 days. Lifespan assays of compound-treated nematodes were performed blinded.

## Carbohydrate determination/phenol–sulfuric acid method

Total carbohydrates were measured with a Total Carbohydrate Assay Kit (Sigma, MAK104). Young adult nematodes were collected with a COPAS Biosort System and stored at −80°C until use. The pellets were disrupted in assay buffer with a sonicator and spun down in a chilled microcentrifuge at 18,800 *g* for 5 min. Total carbohydrates were measured in the supernatant as per the manufacturer's instructions. Absorbance was measured with a Multiskan FC Plate Reader (Thermo Fisher Scientific) at 490 nm.

Total carbohydrate values were normalized to the animal number per sample.

## Cell culture

All HAP1 cell lines were obtained from Horizon Discovery (Horizon Genomics GmbH, Vienna, Austria). Cells were kept in IMDM (Iscove's modified Dulbecco's medium) supplemented with 10% FBS and 1% penicillin/streptomycin. For maintenance purposes, all cell lines were split 1:20–1:30 every 2–3 days. Before compound treatments, $2 \times 10^5$ cells/well were seeded in six-well plates and left overnight to attach in growth medium. All compounds were dissolved in DMSO (0.2% final concentration in the medium).

## Lipidomic profiling

Nematodes were synchronized, and 4,000–8,000 young adults were sorted with a COPAS Biosort System, collected in pellets, flash-frozen in liquid $N_2$, and stored at $-80°C$. A total of five biological replicates per condition were collected. Creative Proteomics Inc. (USA) performed a high-performance liquid chromatography and mass spectrometry (UPLC-MS) analysis to determine the lipids present in wild-type, *age-1*, *gas-1,* and *age-1; gas-1* mutant nematodes. All profiles were normalized to the number of nematodes per sample. Cutoff for significance was 5% FDR. The volcano plot was generated with an R 3.1.3 (R Core Team, 2015).

## Locomotor activity

Gravid nematodes were treated with hypochlorite solution, and eggs were placed on NGM plates seeded with OP50, in the presence of vehicle (DMSO) or one of the indicated compounds. Adult nematodes were transferred every other day until day 10 when the locomotion test was performed. Specifically, 35 nematodes were transferred manually in one well of a transparent flat-bottom 96-well plate containing 100 μl of M9 supplemented with 6 mg/ml heat-inactivated OP50. The plate was placed into a WMicroTracker (Phylumtech S.A., Argentina), and the locomotor activity was recorded for 1 h with a recording frame of 30 min. In each independent assay, at least 3 wells per condition were measured.

## Microscopy

### Fluorescence recovery after photobleaching (FRAP)

L4 stage nematodes were paralyzed with 25 mM Levamisole (AppliChem) in M9 buffer, mounted on 2% agarose pads on glass slides and closed with coverslips. The FRAP assay was performed on an inverted fully motorized Nikon microscope, in association with a Yokogawa Spinning Disk connected to a back-illuminated EMCCD camera (Andor iXON DU-897, 512 × 512 pixels, 16 bit, 35 frames/s). The setup was equipped with a 100× oil immersion lens and with 488-nm emission lasers. Mitochondrial continuity was determined in body wall muscle of transgenic nematodes overexpressing *myo-3::GFP(mit)*. Bleaching was performed with a 488-nm laser in regions of interest (ROIs) of 4 × 13.33 μm in size. Within the bleached area, mitochondria were selected and the mean fluorescence was quantified over time, while the mean cytoplasmic fluorescence was used for the background subtraction. Images were

obtained every 100 ms for a total of 20 s and analyzed by using Fiji imaging software (Open Source). All values were normalized to the pre-bleached intensity. The mobile fraction M(f) (or percentage of recovery) was calculated from the normalized recovery curves in function of the difference between the photobleached point and the maximum recovery state.

### Stimulated emission depletion (STED) microscopy

Super-resolution microscopy was performed using the Leica TCS SP8 gated STED. This instrument is an inverted microscope (Leica DMI6000 CS) equipped with a 100× oil objective lens and a White Light Laser (WLL). Images were taken with a resolution of about 60 nm (pixel size of 18.93 × 18.93 nm) following the use of a depletion laser at 592 nm. In all the experiments, nematodes were paralyzed with 25 mM Levamisole (AppliChem) in M9 buffer mounted on 2% agarose pads on glass slides and closed with coverslips. FRAP experiments were conducted blind to the respective condition (genotype, treatment, or RNAi) of the nematodes.

## Next-generation Sequencing (NGS)

We used five biological replicates of wt, *gas-1*, and *age-1; gas-1* worms for transcriptomics. RNA was extracted as described above, and RNA with a RIN > 9 (RNA Integrity Number) for each replicate was sent to CeGaT GmBH (Tübingen, Germany) for mRNA next-generation sequencing (NGS). Briefly, approximately 100 ng of total RNA was used for library construction (TruSeq Stranded mRNA Library Prep Kit, Illumina). Sequencing was performed with a HiSeq 4000 High Output Mode Illumina platform (1 × 50 bp). The *P*-value significance was determined using the Bonferroni–Hochberg procedure (FDR), and a 5% FDR was used as a cutoff. Scatter plots and Venn diagrams were generated in R 3.1.3.

## Oxygen consumption rate (OCR)

Nematodes were synchronized via hypochlorite solution treatment. At day 4, they were transferred on NGM plates seeded with heat-inactivated OP50. After 2 h, 75 adults/well were transferred in a Seahorse XF24 cell culture microplate containing 525 μl M9 buffer and incubated at 20°C for 1.5 h to recover. In each independent assay, 3–5 wells were used per strain. OCR was measured in a Seahorse XF24 Analyzer (Agilent Technologies Inc., USA). The final concentration of sodium azide was 20 mM. The OCR values were normalized against the average size of each strain as given by the COPAS Biosort (TOF—time of flight). HAP1 cells were seeded in a Seahorse XF24 cell culture microplate with a density of $18 \times 10^3$ cells/well (or $30 \times 10^3$ cells/well when measurement was performed within the day) and left to attach overnight in regular growth medium. The next morning, cells were treated with DMSO (vehicle) or methylxanthines for 24 h. After treatment, growth medium was replaced with Seahorse XF base medium supplemented with 1 mM sodium pyruvate, 10 mM glucose, and 2 mM L-glutamine (pH 7.4). Before the measurement, cells were equilibrated in a $CO_2$-free incubator at 37°C for 45 min. The final concentrations of oligomycin, FCCP, and rotenone/antimycin were 1, 0.125, and 0.5 μM, respectively. After the measurement, cells were trypsinized and counted and the average number of cells/well was used to normalize OCR values.

### Pharyngeal pumping

Pharyngeal contractions were counted over 1 min in young adults. For each experiment, the number of pharyngeal pumps/minute was measured in 10 individual animals coming from three different plates.

### Phosphoproteomics

Mutant nematodes were synchronized, and 8,000–10,000 young adults were sorted with a COPAS Biosort System, collected in pellets, and stored at −80°C. A total of three biological replicates per condition were collected. DC Biosciences Ltd (Scotland, UK) lysed the samples with bead beating, protein was precipitated with methanol:chloroform, digested, and labeled with a Thermo Fisher Scientific TMT™ Mass Tag Labeling Kit. The dry peptide sample was reconstituted, phospho-enriched, fractionated, and subjected to LC-MS/MS/MS analysis. Raw data were filtered for *Escherichia coli* proteins and normalized. The *P*-value significance was determined using the Bonferroni–Hochberg procedure (FDR). Only proteins dysregulated in *gas-1* versus wt with FDR < 20% and an absolute fold change > 5% most extreme control ratios (i.e., 0.345) were used in the subsequent IPA (see also below). The volcano plots of the phosphoproteome data were generated in R 3.1.3.

### qRT–PCR

Young adult nematodes were collected in pellets and frozen overnight at −80°C. Pellets were then pestle-homogenized in the lysis buffer of the RNA Extraction Kit (RNeasy, Qiagen). RNA was isolated as per the manufacturer's instructions. The RNA concentration was quantified with a NanoDrop 2000C (Thermo Fisher Scientific), and 100 ng was used for cDNA synthesis with the qScript Kit (Quantabio) as per the manufacturer's instructions. qRT–PCRs were performed on 1 µl of cDNA with a Fast SYBR Green Master Mix (Applied Biosystems) on an Applied Biosystems qRT–PCR cycler. Each set of primers was verified through a two-step protocol that included (i) testing of nine combinations of primer concentrations against 10 ng of wt cDNA, choosing the combination which gave the lowest $C_T$ value and only the expected product when the qPCR was run on a 2% agarose gel, and (ii) the linearity of $C_T$ values given by the primer combination identified in step (i) with a serial dilution of cDNA ranging from 10 to 0.31 ng. The following primers were used in this study (used concentration indicated in brackets):
*aagr-1* forward: 5′CTTGGTGGGCTGGAGAATTT3′ (200 nM);
*aagr-1* reverse: 5′TCCATCGAAGGGTAGGGTTT3′ (400 nM);
*aagr-2* forward: 5′TCACTACTGGAGACGACTGG3′ (200 nM);
*aagr-2* reverse: 5′TAAGTGGCTTCAATGGCTGG3′ (100 nM);
*aak-2* forward: 5′TTAGCGGAAAGTTGTACGCA3′ (200 nM);
*aak-2* reverse: 5′GCAGGGTTCCACAAAGAAGT3′ (200 nM);
*fzo-1* forward: 5′CCCTGCTCTTGTCAATGATTT3′ (200 nM);
*fzo-1* reverse: 5′AGCAAATTGGTGTTGATTGC3′ (200 nM);
*kin-1* forward: 5′GAAGGACAACAAGAACTCGGC3′ (200 nM);
*kin-1* reverse: 5′GAATGATCCGGTTCCAAGG3′ (400 nM);
*wah-1* forward: 5′GCTGATGCTGTCGAGGAGA3′ (400 nM);
*wah-1* reverse: 5′TGGTGGTGTTCTCTTCTGTAGA3′ (400 nM);

*xdh-1* forward: 5′TCATGAGATGCTCCATTGGT3′ (400 nM);
*xdh-1* reverse: 5′GGAGATGCTGTTGCAATGTT3′ (400 nM);
*actin* forward: 5′TGTGATGCCAGATCTTCTCCAT3′ (100 nM);
*actin* reverse: 5′GAGCACGGTATCGTCACCAA3′ (200 nM);
*pck-1* F: 5′TCCACGTCCAGTTAAGCAAAA3′ (200nM);
*pck-1* R: 5′TGGACGATGGGCGATCATTA3′ (200nM);
*pck-2* F: 5′CATGTTCCGATTCTCAAGGGAG3′ (200nM);
*pck-2* R: 5′ATGTTGAGATCCGTCGCAGA3′ (200nM).

### Statistics

All statistical analyses were performed with GraphPad Prism Software (San Diego, USA), apart from the Benjamini–Hochberg procedure that was used for large datasets (transcriptomics, lipidomics and metabolomics) and was calculated within R 3.1.3.

### STRING and Ingenuity Pathway Analysis (IPA)

STRING analysis (©STRING Consortium 2018, Open Source) was performed on *C. elegans* proteins, which met the FDR and $\log_2$(fold change) cutoffs appropriate for each analysis. For Ingenuity® Pathway Analysis (IPA®), a list of closest human homologs was generated for *C. elegans* genes based on amino acid sequence homology as determined by BLASTP (NCBI). Fold change and *q*-values for the corresponding genes were then imported into IPA (Qiagen, version 01-07) and Core Analysis was performed. FDR and $\log_2$(fold change) cutoffs were selected based on the analysis analyzed, except for NGS in which the FDR was set at 1% to reduce the large number of analysis-ready molecules, as recommended by IPA. For significantly overrepresented canonical pathways and biological functions, -omics data were overlaid with the Molecule Activity Predictor (MAP) function and schematic representations were generated with the Path Designer function.

### Targeted metabolic profiling

Nematodes were synchronized, and young adults were collected in pellets of 250–500 mg and stored at −80°C. Chenomx Inc. (Canada) performed an NMR-1D analysis to determine the metabolites present in wild-type, *gas-1,* and *age-1; gas-1* mutant nematodes. NMR spectra were acquired on a Varian two-channel VNMRS 600 MHz NMR spectrometer equipped with an HX 5 mm probe. All NMR profiles were dilution-corrected to reflect the composition of the original sample. Cutoffs at 10% FDR and |$\log_2$(fold change)| > 1.23 were used.

### TMT/iTRAQ scan

Mutant nematodes were synchronized, and 1,000 young adults were sorted with a COPAS Biosort System, collected in pellets, flash-frozen in liquid $N_2$, and stored at −80°C. A total of three biological replicates per condition were collected. DC Biosciences Ltd (Scotland, UK) lysed the samples with bead beating, and protein was precipitated with methanol:chloroform, digested, and labeled with a Thermo Fisher Scientific TMT™ Mass Tag Labeling Kit. The dry peptide sample was reconstituted, fractionated, and subjected to LC-MS/MS/MS analysis. Raw data were filtered for *E. coli* proteins and normalized. *P*-value significance was determined using the

Bonferroni–Hochberg procedure (FDR), and 20% FDR and 1% ratio of control ratios (i.e., $|\log_2(\text{fold change})| > 0.47$) were used as cutoffs. The volcano plot of the TMT/iTRAQ scan data was generated in R 3.1.3.

### Western blot analysis

Nematode and cell pellets were disrupted via sonication in RIPA buffer (Sigma) supplemented with protease and phosphatase inhibitors (Roche). All lysates were centrifuged, and protein concentration was determined using a Bradford assay (Sigma). Lysates were mixed with Laemmli buffer, and 20 μg was loaded on a 6–12% SDS gel for electrophoresis. Protein transfer on a nitrocellulose membrane was done with a Trans-Blot Turbo™ device (Bio-Rad). All antibodies were used as per the manufacturer's instructions. Immunoblots were visualized with a ChemiDoc imaging system (Bio-Rad).

### Xanthine oxidase activity measurement in nematodes

Xanthine oxidase activity was measured with a Xanthine Oxidase Activity Assay Kit (Sigma, MAK078). Young adult nematodes were collected with a COPAS Biosort System in pellets and stored at −80°C until use. The pellets were disrupted in xanthine oxidase assay buffer with a sonicator and spun down in a chilled microcentrifuge at 14,000 rpm for 10 min. Xanthine oxidase activity was measured in the supernatants as per the manufacturer's instructions. Absorbance was measured with a Multiskan FC Plate Reader (Thermo Fisher Scientific) at 570 nm. Enzymatic activity values were normalized to the animal number per sample.

## Data availability

RNA sequencing has been deposited to the NCBI GEO database under the accession number GSE122902 (https://www.ncbi.nlm.nih.gov/geo/query/acc.cgi?acc=GSE122902). Metabolomics and lipidomics data have been deposited to the EMBL-EBI MetaboLights database (https://doi.org/10.1093/nar/gks1004) with the identifier MTBLS794. The mass spectrometry proteomics and phosphoproteomics data have been deposited to the ProteomeXchange Consortium via the PRIDE partner repository with the dataset identifier PXD011851 (https://www.ebi.ac.uk/pride/archive/projects/PXD011851) and PXD011859 (https://www.ebi.ac.uk/pride/archive/projects/PXD011859), respectively.

**Expanded View** for this article is available online.

## Acknowledgements

Some strains were provided by the CGC, which is funded by NIH Office of Research Infrastructure Programs (P40 OD010440). This research was supported by the DZNE institutional budget and Helmholtz cross-program initiatives "Metabolic Dysfunction" and "Aging and Metabolic Programming (AMPro)". This project was also supported in Germany through the "Bundesministerium für Bildung und Forschung" (BMBF) under the aegis of the EU Joint Programme-Neurodegenerative Disease Research (JPND-www.jpnd.eu). This project has received funding from the European Union's Horizon 2020 research and innovation program under the Marie Skłodowska-Curie grant agreement No 676144 (Synaptic Dysfunction in Alzheimer's Disease, SyDAD). PN, DB, AP, and LW are members of the DFG Cluster of Excellence ImmunoSensation. We would like to kindly acknowledge Mrs. Andrea Linke, Mrs. Christiane Bartling-Kirsch, and Dr. Dagmar Sonntag-Bensch for their technical support. We also express our gratitude to Mr. Chris Gioran for programming the scripts used in the cross-reference of the nematode genes with human homologs for the IPA of the transcriptomic data.

## Author contributions

Conceptualization: AG, AP, and DB; Validation: AG, AP, FB, and JS; Formal Analysis: AG, AP, FB, and LW; Investigation: AG, AP, FB, JS, and LW; Writing: AG and DB; Visualization: AG and DB; Supervision: AG and DB; Project Administration: DB; Funding Acquisition: PN and DB.

## Conflict of interest

The authors declare that they have no conflict of interest.

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
