## [Review Process File · The EMBO Journal]

Multi-omics identify xanthine as a pro-survival metabolite for nematodes with mitochondrial dysfunction

Anna Gioran, Antonia Piazzesi, Fabio Bertan, Jonas Schroer, Lena Wischhof, Pierluigi Nicotera, and Daniele Bano

Review timeline:

Submission date:	4th Apr 2018
Editorial Decision:	30th May 2018
Revision received:	26th Oct 2018
Editorial Decision:	5th Dec 2018
Revision received:	10th Dec 2018
Accepted:	18th Jan 2019

Editor: Elisabetta Argenzio

Transaction Report:

1st Editorial Decision

30th May 2018

Thank you for submitting your manuscript on a role for xanthine derivatives in extending life span of mitochondrial deficient nematodes to The EMBO Journal. Please accept my apologies for the delay in getting back to you due to my travel commitments and detailed discussions in the team. Your study has been sent to two referees for evaluation, and we have received reports from all of them, which are enclosed below. In light of these reports, I am afraid we have concluded that we cannot offer publication here.

As you will see, while the referees consider the findings to be potentially interesting, they also raise major criticism on the study, which in our view precludes going further with this study. In particular, referees #1 finds that nematode and mammalian data are not adequately connected and requires to investigate the role of daf-16 depletion in gas-1 knockout animals. Referee #2 states significant concerns about the lack of important controls in support of the main findings with knockout animals. Also, s/he suggests that independent gas-1 alleles should be used to confirm the findings. Finally, this referee finds that the in vitro versus in vivo effects of xanthine derivatives should be further discussed.

Given these negative opinions from trusted experts in the field and the large number of new additional experiments required, as well as our policy of allowing one major round of revisions only, I am afraid that we cannot offer to invite a revised version of your manuscript at this stage. However, taking into account the potential interest of your findings, I would be willing to look at the study as a new submission at a later time if the referees' concerns could be fully addressed and their suggestions implemented. I should add that for resubmissions we consider novelty at time of submission and, if needed, might involve new referee(s).

I thank you for the opportunity to consider this manuscript. I am sorry that I cannot communicate more positive news, but nevertheless I hope that you will find our referees' comments helpful for improvement of your manuscript.

REFeree REPORTS:

Referee #1:

In this manuscript, authors observed xanthine accumulation as a metabolic product of insulin signaling inhibition in *c. elegans*. They also observed increased survival and fitness in respiratory deficient nematodes upon supplementation of xanthine derivatives. These are new and significant data, which can influence a broad community, especially in the field of mitochondrial disorders. However, the manuscript needs thorough revision to improve the clarity of writing and needs experimental work to strengthen the conclusions.

Major points:

1. In some parts the manuscript is difficult to follow. We suggest that authors explain better the use of certain KO models. For example, what is the cause of impaired OXPHOS in case of *gas-1* KO? why did authors use three HAP1 cell lines with complex I mutations? Which were these mutations?
2. Authors should connect the two aspects of their story (nematode and mammalian cells) experimentally. For example, they shall perform western blots confirming phosphorylation of Akt and PRAS40 analogs in nematodes (or use genetics to test the role of Akt phosphorylation) and measure xanthine levels in NDUFA9 cell line with affected insulin signaling.
3. In Fig1b, a quantification of the western blot is needed. Why mutation in NDUFS2 (a core complex I protein) does have almost no effect on PRAS40 phosphorylation whereas mutations in NDUFA9 and NDUFS4 do? Was this mutation connected with a mild or a severe phenotype? From the Seahorse data it seems to be as serious as NDUFA9 mutant.
4. In Fig1f, authors measured nematode lifespan using *daf-16*KO in combination with upstream protein *age-1* KO and then in triple KO with *gas-1*. If *daf16* KO (on the transcription initiation level) alone can increase the lifespan, what would be its role in *gas-1* KO? This double KO experiment would further confirm the role of insulin signaling pathway.
5. The authors should show the results from qPCR validation of siRNA assays used in Fig3.

Minor points:

1. In Fig1a, it is not clear at which phase of the Seahorse experiment the significance was calculated. Was it basal respiration or maximal respiration?
2. Schematic pictures in Fig.3A, 3B and 4C would deserve deeper explanation in text or at least in figure legend.
3. Fig.3F - "In line with the prediction of the IPA analysis, we found that *age-1*; *gas-1* double mutants displayed an increased PKA/KIN-1/2 activity (Figure 3F)." Compared to what? It is not clear what authors are comparing. Probably double knock-out to *age-1* KO?
4. In Fig.4C - authors should explain why different shades of green and blue were used
5. To better understand the qPCR data it would be beneficial to add efficacy levels for the qPCR primers used.
6. Authors shall place their data in the context of literature, where xanthine oxidase inhibitors were used to treat mitochondrial disorders (e.g. like in PMID 17074601).
7. Do ATP levels change in their complex I mutants treated with PDE inhibitors?

Referee #2:

In this manuscript, Bano and colleagues describe their analysis of the increased lifespan of *gas-1* mutants through the loss of *age-1*, which encodes a PI3 kinase required for insulin signaling. They show that the loss of *age-1* causes metabolic rewiring that results in the preservation of cellular ATP levels. In addition, they identify xanthine accumulation as an effector of a block in insulin signaling and present data in support of the notion that supplementation of xanthine derivatives ameliorates the fitness and survival of *gas-1* mutants.

The role of mitochondrial function or 'health' in the determination of lifespan is still not fully understood. The authors use a defect in insulin signaling, which causes lifespan extension of animals with compromised mitochondrial function (but also of animals carrying other defects), as a tool to functionally understand the mechanism(s) through which lifespan is increased in this context. Unfortunately, in the current manuscript, important controls (wild-type, *age-1* alone) are missing and because of that, the authors' conclusion that this is specific to a defect in mitochondria (rather than an *age-1*-specific effect) is not warranted.

Major points:

1. In many experiments throughout the manuscript, the authors compare *gas-1* and *age-1*; *gas-1* animals without showing the comparison between *age-1* and wild-type animals (experiments presented in parts of Fig 1, 2, 3 and 4). The authors are basically looking at the effects of the loss of *age-1* in a mitochondria-compromised background but they never check whether the effect of the loss of *age-1* is not the same in an otherwise wt background. Similarly, in a number of experiments, the authors do not present data of all the genotypes that would be necessary to conclude from the results. For example, in Fig. 1L, the authors should show *fzo-1*(RNAi) alone as well and not only present the data in Table S1. Similarly, in Fig. 2C, *ife-2* alone is missing, in Fig 3 D, *age-1* and *aak-2* alone and only with *gas-2* should be shown, in Fig. 3 E, *AAK-2* O/E alone is missing etc This is important since at this point, the authors do not have sufficient data in support of the notion that 'Xanthine derivatives promote survival of mitochondrial deficient animals'. One more example: In the text, the authors mention: "We found that IIS inhibition promoted AMPK/AAK-2 activity in complex I deficient animals". However from Figure 3D, it seems that the activation is already present in *age-1* mutants alone. That example shows that *age-1* alone has already a strong effect, independently of the presence of any mitochondrial deficiency.
2. Role of *gas-1*. Throughout the manuscript, the authors use one allele of *gas-1*, *fc21*. The authors should state what kind of mutation *fc21* is. In addition, can they reproduce their findings with an independent *gas-1* allele? Alternatively, can they rescue the *gas-1* phenotypes observed with a *gas-1* rescuing construct?
3. Lifespan assays. It is unclear why there are no error bars on the lifespan data shown. In addition, the authors should clarify how they staged 4 Day adults (did they stage at the L4 stage?). Where the animals scored double-blind? And where the strains used isogenic? Furthermore, the number of animals differ between experiments. For example, in Fig 4E, the numbers range from 62-185 (n=1-2) but in Fig 4G they range from 190-651 (n=2-5). For this reason, especially in Fig 4 E and G, I am not convinced of the difference between wt and *gas-1*. Did the authors perform the experiments presented for *gas-1* in parts F and H also for wild-type? This should be done.
4. Use of the term 'mitochondria deficient animals'. *Gas-1* animals are not really deficient in mitochondria but their mitochondria are slightly compromised with respect to ETC function. This should be clarified throughout the ms.
5. Figure 1. On page 5 of the manuscript, the authors state 'Notably, the lifespan-extending effect of IIS inhibition was not limited to complex I deficient nematodes, since *age-1*(*hx546*); *mev-1*(*kn-1*) double mutants also lived significantly longer compared to complex II deficient *mev-1*(*kn1*) animals (Fig. 1G and Table S1).' However, while the loss of *gas-1* increases lifespan in *age-1*(*lf*) animals, the loss of *mev-1* dramatically decreases lifespan in *age-1*(*lf*) animals. Therefore, while the loss of *age-1* indeed increases lifespan in both mutants, the effects are qualitatively and quantitatively very different especially when one compares the lifespan to that of wild-type and the single mutants. This needs to be addressed.
6. The Authors should check if the *fzo-1*(RNAi) really cause mitochondria fragmentation in *age-1*; *gas-1* animals. The absence of an effect on lifespan observed in Fig 1L could be due to inefficient RNAi knock-down.
7. In Figure 2H, the difference in amount of carbohydrates between *gas-1* and *age-1*; *gas-1* mutant although significant does not seem very strong. The authors should probably mention this in the text. Furthermore, the authors should also check the level of glucose in *age-1* single mutant.
8. In Figure 2F, 3A and 3B, the authors should explain more in the legend and/or in the key in the figure what the different shapes, colors, arrows mean, as it is currently difficult to understand. For example, in figure 3A/B, there are some dotted arrow but it is unclear what that means.
9. In Figure 3, the authors show that *aak-2* RNAi and *kin1* RNAi suppress the long-lived phenotype of *age-1*; *gas1* double mutant. Is this effect specific of the double mutant? The Authors should test whether *aak-2* RNAi and *kin1* RNAi suppress the long-lived phenotype of *age-1* single mutant.
10. In figure S2 B and C, some quantification would be helpful to see how much xanthine derivative decrease phosphorylation of AKT/mTor target and increase PKA activity, especially in comparison to the other drugs used (such as cilostazol, MSX-3...)
11. One caveat that the Authors should mention is that the experiments to establish pleiotropic effects of xanthine derivative were performed in cell culture. Therefore, one cannot exclude that in worms, the effect of these molecules may be different (for example, in term of their effect on PDE or adenosine receptor). This should be mentioned in the discussion.

Minor points:

- 1- In Figure 1 legend, *NDUFS2* and *NDUFS4* are not mentioned.
- 2- Method to measure carbohydrate with phenol sulfuric assay seems missing.

- 3- What is the lower band in g; aOE animals in Figure 3E.
- 4- What is OE construct? Is it uthEx299?
- 5- Typo in Figure 3C: ag;g is probably a;g

We are grateful to the editor and the reviewers for their valuable suggestions. All comments have been taken in consideration and additional experimental evidence has been provided. To accommodate the reviewers' inputs and call better the attention to the results, we have partly re-organized the structure and re-worded some concepts in our manuscript. We believe that the quality of our work has greatly improved and our conclusions are further supported by the new findings. The point-by-point response to the reviewers is below.

Reviewers' comments and point-by-point response:

Reviewer #1.

*In this manuscript, authors observed xanthine accumulation as a metabolic product of insulin signaling inhibition in *C. elegans*. They also observed increased survival and fitness in respiratory deficient nematodes upon supplementation of xanthine derivatives. These are new and significant data, which can influence a broad community, especially in the field of mitochondrial disorders.*

We would like to thank the reviewer for such a kind comment.

However, the manuscript needs thorough revision to improve the clarity of writing and needs experimental work to strengthen the conclusions.

Major points:

*1. In some parts the manuscript is difficult to follow. We suggest that authors explain better the use of certain KO models. For example, what is the cause of impaired OXPHOS in case of *gas-1* KO?*

The *gas-1* gene encodes for a subunit of the respiratory complex I homologous to the human NADH dehydrogenase iron-sulfur protein 2 (NDUFS2). The *gas-1(fc21)* allele was isolated in a screening of *C. elegans* mutants sensitive to volatile anesthetics (Kayser et al, 1999) and has been extensively used as a model of mitochondrial dysfunction (Kayser et al, 2001; Hartman et al, 2001; Ichishita et al, 2008; Sanz et al, 2009; Troulinaki et al, 2018). As a missense mutation, *gas-1(fc21)* results in the posttranslational loss of complex I subunit NDUFS2 and, as a consequence, aberrant mitochondrial function, diminished fitness and decreased survival (Kayser et al, 2001; Kayser et al, 2004; Dingley et al., 2010; Chang et al., 2015; Koopman et al., 2016; Troulinaki et al, 2018). Following the reviewer's advice, all these details have been included in the Result section.

Why did authors use three HAP1 cell lines with complex I mutations? Which were these mutations?

We used mitochondrial deficient HAP1 cells because we aimed to investigate further some of the molecular processes identified through our comprehensive genetic analysis in nematodes. There different lines were used to verify that the aberrant Akt/mTOR signaling occurs in a mutation-independent manner. One obvious advantage of human cells is the possibility to employ specific reagents (e.g., antibodies) that efficiently work in mammalian samples. As described in the original manuscript, we firstly performed an independent assessment of the Akt/mTOR pathways in these cells carrying different mitochondrial defects. In line with previously published papers (Johnson et al, 2013; Siegmund et al, 2017, Zheng et al, 2016; Ising et al, 2015), we showed that mitochondrial mutant cells exhibited an increased activity of Akt and mTOR (Supplementary Figure S4C), further supporting that hyperactive insulin/IGF-1

signaling (IIS) is a common signature of mitochondrial dysfunction. With this gain of knowledge, we went on and studied the IIS pathway in both mitochondrial deficient cells and nematodes, demonstrating that some of the molecular processes are evolutionarily conserved. Following the reviewer's comment, we have included information relative to the mutations of the different cell lines (in the text and in Supplementary Figure S4A). We excluded the data on NPCs, since they became redundant with the rest of the models.

2. *Authors should connect the two aspects of their story (nematode and mammalian cells) experimentally. For example, they shall perform western blots confirming phosphorylation of Akt and PRAS40 analogs in nematodes (or use genetics to test the role of Akt phosphorylation) and measure xanthine levels in NDUFA9 cell line with affected insulin signaling.*

We thank the reviewer for this question. Unfortunately, only a handful of commercially available antibodies work in *C. elegans*, limiting immunoblot analyses in nematodes. However, we went beyond her/his request and have addressed her/his comment in an unbiased manner by performing an additional phosphoproteomic analysis (Figure 1D-1E and Supplementary Figure S1A-1B). Our set of data demonstrates the following:

- a. approximately 97% of differentially phosphorylated proteins are IIS-dependent, as demonstrated by the fact that they revert back to wt levels in *age-1*; *gas-1* mutants
- b. In *gas-1* mutants, 5 of the 7 predicted hyperactive pathways are interconnected with IIS.

We would like to point out that the submitted phosphoproteome is one of the few performed in nematodes.

To further support our conclusions, we have performed additional epistatic lifespan analyses. Since PDK-1 is required for the full activation of AKT, we generated *pdk-1*; *gas-1* double mutants and performed lifespan assay. In line with our prior data, we found that *pdk-1* loss-of-function extends *gas-1(fc21)* survival (Supplementary Figure S1C).

The reviewer suggested to measure xanthine levels in *NDUFA9* KO cells after downregulating the IIS. Assessment of metabolites in HAP1 cells has been one of our aim in the past. We tried twice to perform metabolomic analyses in cells, however we obtained measurements only for a handful of metabolites. Specifically, we were able to detect Ala, choline, creatine, Glu, Gln, glutathione, glycerol (probably a contaminant), Gly, Ile, lactate, Leu, N-acetylaspartate, o-phosphocholine, taurine, Val. Following the reviewer's comment, we have tried to measure xanthine in WT and *NDUFA9* KO HAP1 cells, where cells were transfected with siRNA against IRS1/2 or PI3K. Despite our experimental effort, we were not able to obtain reliable data and do not feel confident to draw conclusions based on this assessment. We consider the reviewer's comment a valid point, even though we believe that *in vitro* models may not fully recapitulate all processes occurring in a whole organism.

3. *In Fig1b, a quantification of the western blot is needed.*

We have included densitometry of immunoblot analyses (now Supplementary Figure S4C).

Why mutation in NDUFS2 (a core complex I protein) does have almost no effect on PRAS40

phosphorylation whereas mutations in NDUFA9 and NDUF54 do? Was this mutation connected with a mild or a severe phenotype? From the Seahorse data it seems to be as serious as NDUFA9 mutant.

We have performed additional immunoblots and have confirmed the upregulation of pPRAS40/PRAS40 ratio in both *NDUFS2* and *NDUFS4* KO cells. As requested, we have included the densitometry values, showing that the upregulation is consistent throughout different biological replicates. New panels have been generated and included in the revised manuscript. Overall, we feel confident to say that enhanced IIS occurs in all our mitochondrial deficient HAP1 cells. However, we are not in the position to discuss whether one line has higher or lower Akt/mTOR activity. Definitely, this heterogenous behavior is in line with the intrinsic variability of human mitochondrial diseases, in which mutations in the same gene give rise to different pathologies, as well as different mutated genes are linked to the same syndrome (as in the case of Leigh disease).

4. In Fig 1f, authors measured nematode lifespan using daf-16 KO in combination with upstream protein age-1 KO and then in triple KO with gas-1. If daf-16 KO (on the transcription initiation level) alone can increase the lifespan, what would be its role in gas-1 KO? This double KO experiment would further confirm the role of insulin signaling pathway.

DAF-16 KO does not extend the lifespan of *gas-1* mutant nematodes (Supplementary Figure S1D).

5. The authors should show the results from qPCR validation of siRNA assays used in Fig3.

As stated in the Materials and Methods, silencing of target genes was assessed and confirmed by qPCR. In the submitted manuscript, these data have been provided as a Supplementary Figure (S1E, S3A, S3C).

Minor points:

1. In Fig1a, it is not clear at which phase of the Seahorse experiment the significance was calculated. Was it basal respiration or maximal respiration?

The significance was referred to basal respiration. We have adjusted this panel and the corresponding figure legend accordingly.

2. Schematic pictures in Fig. 3A, 3B and 4C would deserve deeper explanation in text or at least in figure legend.

In the revised manuscript, we have given more details in the figure legends.

3. Fig.3F - "In line with the prediction of the IPA analysis, we found that age-1; gas-1 double mutants displayed an increased PKA/KIN-1/2 activity (Figure 3F)." Compared to what? It is not clear what authors are comparing. Probably double knock-out to age-1 KO?

Following the reviewer's comment, we have corrected the sentence to "we found that age-1;

gas-1 double mutants displayed an increased KIN-1 activity compared to gas-1 mutant animals”.

4. *In Fig. 4C - authors should explain why different shades of green and blue were used*

We have improved the figure legend accordingly.

5. *To better understand the qPCR data it would be beneficial to add efficacy levels for the qPCR primers used.*

To address this point, in the Materials and Methods we have added a paragraph on how qPCR primers were validated and at which concentration they were used for each experiment.

6. *Authors shall place their data in the context of literature, where xanthine oxidase inhibitors were used to treat mitochondrial disorders (e.g. like in PMID 17074601).*

In the Results and Discussion sections, we have included a few sentences summarizing what is generally known on xanthine metabolism. Moreover, we discussed our findings in light of what has been previously published on xanthine derivatives. We would like to highlight that methylxanthines do not inhibit xanthine oxidase in our settings, since our assessments indicate that xanthine oxidase activity remains unchanged in *age-1*; *gas-1* nematodes compared to *gas-1* mutants.

7. *Do ATP levels change in their complex I mutants treated with PDE inhibitors?*

Following the reviewer's comment, we have measured ATP levels in *gas-1* mutant nematodes treated with the xanthine derivatives, however did not observe any significant change compared to untreated animals (data not shown). We believe that this is due to the milder effect of methylxanthines compared to genetic inhibition of IIS. Conversely, we observed a drastic decrease of ATP levels in AAK-2 or KIN-1 deficient *age-1*; *gas-1* mutant animals (Figure 3H-3I). This set of data supports further the importance of these two kinases in the regulation of energy-saving pathways.

Referee #2:

In this manuscript, Bano and colleagues describe their analysis of the increased lifespan of gas-1 mutants through the loss of age-1, which encodes a PI3 kinase required for insulin signaling. They show that the loss of age-1 causes metabolic rewiring that results in the preservation of cellular ATP levels. In addition, they identify xanthine accumulation as an effector of a block in insulin signaling and present data in support of the notion that supplementation of xanthine derivatives ameliorates the fitness and survival of gas-1 mutants. The role of mitochondrial function or 'health' in the determination of lifespan is still not fully understood. The authors use a defect in insulin signaling, which causes lifespan extension of animals with compromised mitochondrial function (but also of animals carrying other defects), as a tool to functionally understand the mechanism(s) through which lifespan is increased in this context. Unfortunately, in the current manuscript, important controls (wild-type, age-1

alone) are missing and because of that, the authors' conclusion that this is specific to a defect in mitochondria (rather than an *age-1*-specific effect) is not warranted.

Major points:

1. In many experiments throughout the manuscript, the authors compare *gas-1* and *age-1*; *gas-1* animals without showing the comparison between *age-1* and wild-type animals (experiments presented in parts of Fig 1, 2, 3 and 4). The authors are basically looking at the effects of the loss of *age-1* in a mitochondria-compromised background but they never check whether the effect of the loss of *age-1* is not the same in an otherwise wt background. Similarly, in a number of experiments, the authors do not present data of all the genotypes that would be necessary to conclude from the results.

Although we adopted a standard graphical representation of our data generally accepted in the field, we have added the missing lifespans as suggested by the reviewer (Figure 1K and Supplementary Figure S1I, S3B, S3D). We would kindly point out that most of these data were already available in the manuscript, with all the statistics distinctly reported in Table S1.

For example, in Fig. 1L, the authors should show *fzo-1*(RNAi) alone as well and not only present the data in Table S1.

We have substituted the panel with one showing also *fzo-1* silenced wt nematodes (Figure 1K and Supplementary Figure S1I). Moreover, we have provided additional representative images of the mitochondrial network of *age-1*; *gas-1* mutants grown on *fzo-1* RNA (Supplementary Figure S1G). To confirm further the efficient downregulation of *fzo-1* expression, we have added a RT-PCR analysis performed on mRNA extracted from wt animals grown on RNAi expressing bacteria (Supplementary Figure S1H).

Similarly, in Fig. 2C, *ife-2* alone is missing, in Fig 3 D, *age-1* and *aak-2* alone and only with *gas-2* should be shown, in Fig. 3 E, *AAK-2* O/E alone is missing etc. This is important since at this point, the authors do not have sufficient data in support of the notion that 'Xanthine derivatives promote survival of mitochondrial deficient animals'.

We have substituted those panels with new graphs containing also the lifespans of single mutants (Figure 2C, 3D-3E, Supplementary Figure S3B).

One more example: In the text, the authors mention: "We found that IIS inhibition promoted AMPK/AAK-2 activity in complex I deficient animals". However, from Figure 3D, it seems that the activation is already present in *age-1* mutants alone. That example shows that *age-1* alone has already a strong effect, independently of the presence of any mitochondrial deficiency.

That is correct and in line with one of our previously published manuscripts (Gioran et al, 2014). Inhibition of the IIS induces the activity of AMPK/AAK2, which is even more enhanced in mitochondrial deficient animals. Following the reviewer's comment, the new sentence reads: "In line with its role in conditions of low energy and nutrient deprivation, and with our IPA predictions, we found that *age-1*; *gas-1* double mutants had increased levels of phospho-AAK-2 compared to *gas-1* animals (Figure 3C) as revealed using a previously validated antibody".

2. *Role of gas-1. Throughout the manuscript, the authors use one allele of gas-1, fc21. The authors should state what kind of mutation fc21 is. In addition, can they reproduce their findings with an independent gas-1 allele? Alternatively, can they rescue the gas-1 phenotypes observed with a gas-1 rescuing construct?*

The *gas-1* gene encodes for a subunit of respiratory complex I homologous to the human NADH dehydrogenase iron-sulfur protein 2 (NDUFS2). The *gas-1(fc21)* allele is a missense mutation (R290K), which results in the posttranslational loss of complex I subunit NDUFS2 and, as a consequence, aberrant mitochondrial function, diminished fitness and decreased survival (Kayser et al, 2001; Troulinaki et al, 2018).

There are no other alleles for the *gas-1* gene. For the past two years, we have been trying to obtain another *gas-1* mutant. In collaboration with a company (Knudra/NemaMetrix), we also tried a gene editing (i.e., CRISPR/Cas9 method) of the *gas-1* gene. The idea was to substitute all introns and exons with a cassette encoding for GFP. While we obtained heterozygous mutant animals, we were not able to generate homozygous mutants, suggesting that *gas-1* KO animals are lethal at the embryonic stage.

The rescue of *gas-1(fc21)* mutation has been already performed and characterized (Kayser et al, 2001).

We would like to point out that, although we used only one *gas-1* mutation, we were able to recapitulate the lifespan-extending properties of IIS inhibition also in complex II mutant (*mev-1*) animals, untangling the effect of reduced IIS inhibition from distinct mitochondrial defects (Figure 1F). Furthermore, we have included a set of data showing that IIS inhibition significantly extends the survival of the short-lived WAH-1/AIF deficient animals (Supplementary Figure S1F). Taken together, our data suggests that reduced IIS counteract OXPHOS deficiency in a large array of mitochondrial mutant models.

3. *Lifespan assays. It is unclear why there are no error bars on the lifespan data shown. In addition, the authors should clarify how they staged 4 days adults (did they stage at the L4 stage?). Where the animals scored double-blind? And where the strains used isogenic?*

This is a standard graphical representation of lifespan data generally accepted in the field. Each curve is one biological replicate of the respective condition. As in one of our previous papers (Piazzesi et al, 2016), we have added under each plot the average median survival \pm SEM of each condition as calculated across all of the biological replicates and as summarized in Table S1.

In the Materials and Methods we stated that “*synchronized day 4 adults were transferred to fresh NGM plates...*”. We have corrected the sentence by adding “*synchronized day 4 (since hatching) young adults were transferred...*”.

Wherever possible, we performed lifespans blinded as indicated in the text.

All nematode populations used in our experiments were isogenic by definition.

Furthermore, the number of animals differs between experiments. For example, in Fig 4E, the numbers range from 62-185 (n=1-2) but in Fig 4G they range from 190-651 (n=2-5). For this reason, especially in Fig 4 E and G, I am not convinced of the difference between wt and gas-1. Did the authors perform the experiments presented for gas-1 in parts F and H also for wild-type? This should be done.

For all our lifespan experiments, we followed accepted protocols and recommended procedures (as an example: Sutphin et al, 2009). The number of starting animals depends on the strains, since some of them are very sensitive and more susceptible to censoring (i.e., internal egg hatching, vulva protrusion etc). For some of the experiments, we have included additional biological replicates.

We have performed pharyngeal pumping assays for wt nematodes treated with methylxanthines as part of our initial characterization of the compounds' effect. We found no differences compared to vehicle-treated animals. See the data below.

4. Use of the term 'mitochondria deficient animals'. gas-1 animals are not really deficient in mitochondria but their mitochondria are slightly compromised with respect to ETC function. This should be clarified throughout the ms.

It may be semantic, but “deficiency is the quality or state of being defective or of lacking some necessary quality or element”. We believe that we and others correctly describe *gas-1* mutants as “mitochondria(1) deficient animals”, since nematodes carrying the *gas-1(fc21)* allele exhibit a severe complex I defect. In line with this argument, our group and others showed that complex I activity is significantly diminished in *gas-1(fc21)* mutants compared to wt (Kayser et al, 2001; Pujol et al, 2013; Troulinaki et al, 2018). In the submitted manuscript, we have reported that the mitochondrial respiration and the ATP levels of these mutants are reduced by approximately half, compared to wt (Figure 1D-1I-1H). Following the reviewer’s comment, we have added additional explanation about the nature of mitochondrial dysfunction in *gas-1(fc21)* mutant nematodes.

5. Figure 1. On page 5 of the manuscript, the authors state 'Notably, the lifespan-extending effect of IIS inhibition was not limited to complex I deficient nematodes, since age-1(hx546); mev-1(kn-1) double mutants also lived significantly longer compared to complex II deficient mev-1(kn1) animals (Fig. 1G and Table S1).' However, while the loss of gas-1 increases lifespan in age-1(lf) animals, the loss of mev-1 dramatically decreases lifespan in age-1(lf) animals. Therefore, while the loss of age-1 indeed increases lifespan in both mutants, the

effects are qualitatively and quantitatively very different especially when one compares the lifespan to that of wild-type and the single mutants. This needs to be addressed.

We thank the reviewer for this comment. As written in the manuscript, the most prominent hallmark of mitochondrial diseases is altered mitochondrial function and diminished oxidative phosphorylation (OXPHOS). Mutations may occur in a sporadic or inherited fashion and the resulting clinical manifestations show a striking variability in terms of age onset, organ involvement, symptoms, disease progression and lifespan expectancy. Due to the genetic and structural complexity of the OXPHOS system, the loose genotype-phenotype correlation complicates diagnostics as well as the development of effective treatments. As in mammals, the survival of mitochondrial mutant nematodes is different according to the mutations. We have no doubt that the reviewer is aware that some of these mitochondrial defects reduce lifespan (e.g., *gas-1* and *mev-1*), while others promote longevity (e.g., *nuo-6* and *clk-1*) in nematodes. During our study, we noticed that *age-1; mev-1* mutants have a shorter lifespan compared to *age-1; gas-1* mutants. Our explanation is that *mev-1* animals are sicker compared to *gas-1* mutants. In fact, *mev-1* animals have a median lifespan of 18 ± 1 whereas *gas-1* mutants live 21.6 ± 0.7 days (see Table S1). Moreover, *mev-1* mutants are more frequently censored during a lifespan assay (1212 censored out of 1469 in total) compared to *gas-1* (927 censored out of 1643 in total; see Table S1), indicating a more stressed phenotype. Overall, we are not surprised that the *age-1* mutation cannot stimulate survival of *mev-1* mutants as much as for *gas-1* mutants. Having that said, our epistatic analysis demonstrates that *age-1; mev-1* double mutants live significantly longer than *mev-1* single mutants, while their median lifespan is indistinguishable from that of wt animals (Figure 1G). In the revised manuscript, we have added a few lines about this biological aspect. Moreover, we have added additional lifespan assay using other short-lived mitochondrial mutants (i.e., *wah-1* silenced wt animals).

6. The Authors should check if the fzo-1(RNAi) really cause mitochondria fragmentation in age-1; gas-1 animals. The absence of an effect on lifespan observed in Fig 1L could be due to inefficient RNAi knock-down.

The effect of *fzo-1* (RNAi) on mitochondrial dynamics is reported in Figure 1I-1J and Supplementary Figure S1G. In Figure 1J, we performed FRAP (Fluorescence Recovery After Photobleaching) measurements in *age-1; gas-1* double mutants upon *fzo-1* RNAi. We showed that *fzo-1* RNAi (from hatching until the L4 stage) is sufficient to promote mitochondrial fragmentation in *age-1; gas-1* mutants. Additional representative images of the mitochondrial network of *age-1; gas-1* animals upon *fzo-1* RNAi have been provided (Supplementary Figure S1G). Furthermore, we have added qRT-PCR of *fzo-1* downregulation (Supplementary Figure S1H).

7. In Figure 2H, the difference in amount of carbohydrates between gas-1 and age-1;gas-1 mutant although significant does not seem very strong. The authors should probably mention this in the text. Furthermore, the authors should also check the level of glucose in age-1 single mutant.

To support further our conclusions, we have performed glucose and carbohydrate measurements in wt, *age-1*, *gas-1*, *age-1; gas-1* animals (Figure 2H). Moreover, we have

provided additional evidence on expression levels of proteins and genes involved in carbohydrate metabolism in the aforementioned strains (Supplementary Figure S2B-2C and Table S2). This set of data should further support the effect of IIS inhibition on this distinct metabolic process and should be sufficient to address the reviewer's comment.

8. In Figure 2F, 3A and 3B, the authors should explain more in the legend and/or in the key in the figure what the different shapes, colors, arrows mean, as it is currently difficult to understand. For example, in figure 3A/B, there are some dotted arrows but it is unclear what that means.

In the revised manuscript, we have added a more thorough description in the figure legend.

9. In Figure 3, the authors show that aak-2 RNAi and kin1 RNAi suppress the long-lived phenotype of age-1; gas1 double mutant. Is this effect specific of the double mutant? The Authors should test whether aak-2 RNAi and kin1 RNAi suppress the long-lived phenotype of age-1 single mutant.

As previously reported (Apfeld et al, 2004), *aak-2* loss-of-function reduces the lifespan of IIS mutant nematodes. In line with some of our preliminary data, we have shown that *aak-2* and *kin-1* RNAi reduce the lifespan of *age-1* mutant animals (Supplementary Figure S3B-3D). It is worth noting that in the manuscript we do not claim that the lifespan-extending properties of these kinases are specific to IIS deficient, mitochondrial mutant animals.

10. In figure S2 B and C, some quantification would be helpful to see how much xanthine derivative decrease phosphorylation of AKT/mTOR target and increase PKA activity, especially in comparison to the other drugs used (such as cilostazol, MSX-3...).

To make our results clearer, we have provided densitometry of our western blots.

11. One caveat that the Authors should mention is that the experiments to establish pleiotropic effects of xanthine derivative were performed in cell culture. Therefore, one cannot exclude that in worms, the effect of these molecules may be different (for example, in term of their effect on PDE or adenosine receptor). This should be mentioned in the discussion.

We have discussed further the pleiotropic effects of xanthine derivatives.

Minor points:

1- In Figure 1 legend, NDUFS2 and NDUFS4 are not mentioned.

We have included this information in the revised manuscript.

2- Method to measure carbohydrate with phenol sulfuric assay seems missing.

We apologize for the omission. We have included this information in the revised manuscript.

3- *What is the lower band in g; aOE animals in Figure 3E.*

In this strain, a truncated AAK-2 (aa1-aa321) protein is fused to GFP. The resulting protein is estimated to be 3kDa smaller than the endogenous full AAK-2 (upper band). We have included this information in the figure legend to avoid any misinterpretation.

4- *What is OE construct? Is it uthEx299?*

Correct, it is the uthEx299 extrachromosomal array (Mair et al, 2012).

5- *Typo in Figure 3C: ag;g is probably a;g*

We have corrected the figure accordingly.

Thank you for submitting a revised version of your manuscript and please accept my apologies for the delay in getting back to you with our decision. Your study has now been seen by the original referees, whose comments are shown below.

As you will see, while referee #1 finds that all criticisms have been addressed satisfactorily, referee #2 still points to one issue and requests you to clarify the role or significance of mitochondrial morphology in the context of your findings.

In addition to resolving this remaining point from referee #2, there are a few editorial issues concerning text and figures that I need you to address before we can officially accept the manuscript.

REFeree REPORTS:

Referee #1:

The authors took our comments in careful consideration and run the extra mile to provide the required experimental validations. In particular, the phosphoproteome data considerably strengthen the conclusions and clarify that there is a conserved role from nematode to mammalian cell lines for the supplementation of Xanthine derivatives to suppress the aberrant IIS activation in organisms with mitochondrial dysfunction. This is an appealing therapeutic strategy that is well worth the broad readership of EMBO Journal

Referee #2:

The revised manuscript by Bano and coworkers is significantly improved and many of the concerns of the reviewers have been addressed. There is one thing that the authors should address in a modified way and that is the role or significance of mitochondrial morphology in the context of their findings. This is not addressed in a satisfying manner in the current manuscript.

Corresponding Author Name: Daniele Bano

Journal Submitted to: The EMBO JOURNAL

Manuscript Number: EMBOJ-2018-99558R-Q